# Sliding Window Recurrences for Sequence Models

**Dragos Secrieru**[*]                                                                  *dragos@radicalnumerics.ai*
*Mila, Université de Montréal*
*RIKEN AIP*
*Radical Numerics Inc.*

**Garyk Brixi**[*]                                                                      *gbrixi@stanford.edu*
*Radical Numerics Inc.*
*Stanford University*

**Yoshua Bengio**[†]                                                                  *yoshua.bengio@mila.quebec*
*Mila, Université de Montréal*

**Taiji Suzuki**[†]                                                                    *taiji@mist.i.u-tokyo.ac.jp*
*University of Tokyo*
*RIKEN AIP*

**Michael Poli**[†]                                                                    *michael@radicalnumerics.ai*
*Radical Numerics Inc.*
*Stanford University*

**Stefano Massaroli**[*]                                                               *stefano@radicalnumerics.ai*
*Radical Numerics Inc.*
*RIKEN AIP*

**Reviewed on OpenReview:** *https://openreview.net/forum?id=VO9uO7Oouz*

## Abstract

Multi-hybrid architectures are poised to take over language modeling due to better quality and performance. We introduce a hierarchical decomposition framework for linear recurrences that yields Sliding Window Recurrences, a windowed training mode for recurrence layers in hybrid models. Unlike sliding-window attention, SWR is derived from the transfer structure of recurrences: it truncates the carrier system induced by the decomposition while preserving dense local recurrence dynamics. We focus specifically on hardware-aligned windows which are naturally jagged, limiting costly inter-warp communication. Using SWR, we develop Phalanx layers for hybrid language models. In 1B parameter multi-hybrid models, Phalanx achieves over 10-40% speedup across 4K to 16K context length over optimized Transformers while matching perplexity.

## 1 Introduction

Several sub-quadratic token-mixing primitives deliver high quality in language modeling when combined with Attention, in so-called hybrid models. Some methods achieve efficiency by limiting token-mixing to nearby tokens, such as *sliding-window attention* (SWA)(Beltagy et al., 2020; Agarwal et al., 2025) and gated short convolutions (Ku et al., 2025; Thomas et al., 2024; Chandrasegaran et al., 2025). Meanwhile, others are designed to capture global token interactions such as linear recurrences (Yang et al., 2023; Gu & Dao, 2023; Yang et al., 2024; Arora et al., 2024; Zhang et al., 2024) or gated long convolutions (Poli et al., 2023; Massaroli et al., 2023). The resulting *transfer operator* (the surrogate attention matrix) is typically dense but highly

---

[*]Equal contribution. [†]Equal senior authorship.

structured with decay so that token interactions have exponentially diminishing effect with longer distance, while accruing greater computational cost due to data movement across the GPU memory hierarchy.

In light of the strong quality of hybrids and the central role of data movement on modern GPUs, we ask whether it is possible to retain modeling quality while making *hardware-aligned data locality* a key operator design axis. Our answer are windowed recurrences, realized by algorithms that map well onto the GPU memory hierarchy. Concretely, we contribute:

**Sliding Window Recurrences** We introduce *sliding window recurrences* (SWRs) in Section 4, a family of truncated sequence mixer primitives that makes hardware-aligned data locality an explicit design axis. In practical SWRs the *window* is *jagged* rather than uniform (Figure 1.1, right) to be efficiently realized on GPUs.

**Kernels and numerics.** We implement SWRs with a block-two-pass (B2P) algorithm and kernel that avoids thread-block carry chains (Section 4.2.1). Each warp fuses its local computation into high-throughput GEMMs, and a *single device-wide parallel rank-1 update* broadcasts the residual needed by the next warp. In the causal setting, communication happens only with the immediately preceding warp; the top-level transfer becomes diagonal, achieving logical depth 1 with *one* thread-block synchronization. We cast scan algorithms as *matrix factorizations* of the transfer operator, exposing flat vs. hierarchical decompositions that map one-to-one onto GPU hierarchy. We adopt an aggressive truncation with block size of 16 to match warp size on GPU, maximizing efficiency while still retaining downstream language modeling performance. This band is **8 times** shorter than modern variants of SWA (Agarwal et al., 2025).

**Phalanx layer for hybrid architectures.** We introduce Phalanx, a new layer for short-range token mixing using SWRs in Section 5. Phalanx-Attention hybrid models preserve the quality of SWA/Attention hybrids while training **28% faster at 8K** context length. The Phalanx hybrid is also the fastest even at short lengths, improving end-to-end training throughput by **10%** over pure Attention on Hopper GPUs.

## 2 Related Work

**Local attention and convolutional mixers.** Local attention restricts receptive fields to improve arithmetic intensity and locality. Examples include sliding window attention (Beltagy et al., 2020; Agarwal et al., 2025). Complementary convolutional approaches (including implicit convolutions) likewise emphasize local computation with strong scaling (Romero et al., 2021; Poli et al., 2023; Ku et al., 2025). Our work shares the emphasis on locality but differs by *explicitly band-limiting inter-tile transfer* via a recurrence matched to the GPU hierarchy.

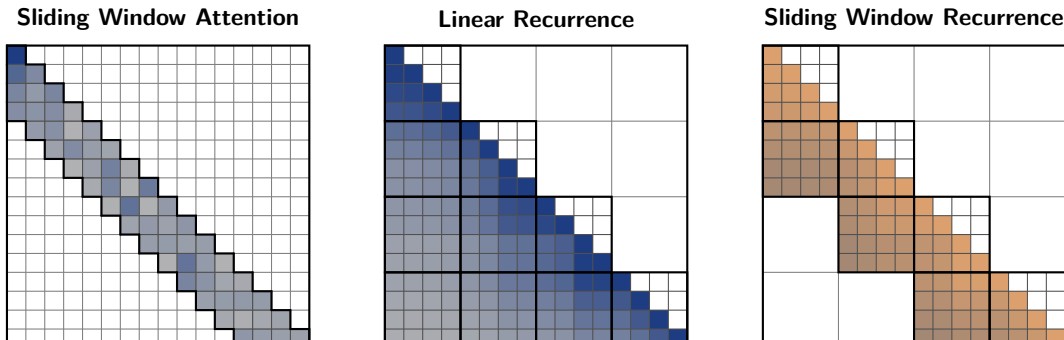

Figure 1.1: **Sliding Window Recurrences (SWR):** A new class of primitives for hardware-aligned sequence mixing. (Left) *sliding window attention.* (Middle) Full *linear recurrence* with exponentially decaying bands. (Right) *sliding window recurrence* (ours) with jagged window structure, computed efficiently via the proposed *Block Two-Pass* (B2P) algorithm. This jagged structure naturally aligns with GPU memory hierarchies, enabling the Phalanx layer to achieve higher end-to-end training throughput while preserving quality at scale.

**Linear recurrences, SSMs, and semiseparable structure.** Recurrence-based layers such as GLA, Mamba, and Gated DeltaNet (Yang et al., 2023; Gu & Dao, 2023; Yang et al., 2024) realize transfer operators that are *sequentially semiseparable*, yielding exponentially decaying off-diagonal bands. This both motivates our finite-precision horizon and informs our band-limited design. Connections between SSMs and attention via structured matrices further clarify algorithmic trade-offs and hardware mapping (Dao & Gu, 2024).

**Hybrid architectures, and hardware-aware design.** Hybrid architectures (e.g., mixing attention with local operators) achieve the best scaling rates (Poli et al., 2024; Wang et al., 2025b). One common option is sliding window attention paired with global attention (Brown et al., 2020; Agarwal et al., 2025) or linear attention (Arora et al., 2024). Modern approaches to hybridization employ a *multi-hybrid* stack of operators with multiple window sizes (Ku et al., 2025). Phalanx fits this direction as a local specialist that delegates global routing to attention.

**Parallel scan and GPU implementations.** Parallel scan underpins fast recurrences. Classic formulations cite Blelloch's prefix-sum (Blelloch, 1990) and the asymptotically efficient Brent-Kung (BK) algorithm (Brent & Kung, 1982). However, BK is *flat*, while GPUs are *hierarchical*; high-performance libraries (CUB/NCCL) implement hierarchical scans and incorporate the *decoupled look-back* strategy to hide inter-block latency (Merrill & Garland, 2016). SWR embraces hierarchy explicitly: at the highest indexing level, the transfer is diagonal (logical depth 1), requiring only a single thread-block synchronization with strictly local inter-warp communication. This jagged structure being more efficiently realized on GPU was previously used for windowed attention variants Zaheer et al. (2020); Zhang et al. (2024).

## 3 A Matrix Theory of Linear Recurrences

We analyze the algebraic structure of first-order linear recurrences, a fundamental computational primitive for signal processing and deep learning alike. Our objective is to develop a matrix-based representation of the operator that maps an input sequence to the corresponding state sequence. This algebraic framework is the foundation for a systematic understanding of the system's dependency structure and for the principled derivation of computational algorithms. Building on this representation, we discuss two complementary algorithmic families. First, *flat* matrix-factorization methods (Section 3.1) factor the transfer operator into logarithmically many sparse factors, yielding parallel algorithms such as Kogge-Stone or Brent-Kung scans. Second, *hierarchical* block algorithms (Section 3.2) tile the sequence, compress inter-block coupling to a scalar carrier (rank-one off-diagonals), and split computation into parallel local solves and a coarse global recurrence; this design matches memory hierarchies and admits efficient matrix-multiplication implementations via numerically stable log-space materialization.

> *Flat vs. hierarchical algorithms.* The categorization of fast matrix multiplication algorithms into *flat* and *hierarchical* is essential in modern high-performance computing. This distinction addresses the widening gap between computational throughput and the cost of data movement. As architectures increasingly rely on deep memory hierarchies (e.g., GPUs), algorithms must be evaluated not just by arithmetic complexity, but by their communication patterns and data locality.
> Professor David Keyes discusses the foundations for a hardware-centric taxonomy of matrix algorithms:
>
> > *Two universes of computational linear algebra exist today side-by-side, a flat, sequential universe in which algorithms are simply stated with loops over global address spaces that typically process a row or a column at a time and another universe in which algorithms are restated for performance in ways that exploit hierarchy, with loops over local ranges only at each hierarchical level.* (Keyes et al., 2020)
>
> *Flat algorithms* achieve efficiency, typically $O(n \log n)$ work, through a factorization of a matrix $\boldsymbol{M}$ into a (short) sequence of sparse factors:
>
> $$\boldsymbol{M} = \boldsymbol{F}_m \boldsymbol{F}_{m-1} \cdots \boldsymbol{F}_1$$

where the depth $m = O(\log n)$, and each factor $\boldsymbol{F}_k$ contains at most $O(n)$ non-zero entries. These algorithms are flat because their data dependency structure often spans the entire input domain. The sparsity pattern involves long-range connections (e.g., the FFT butterfly network), necessitating global scattering or gathering of information. This fails to exploit data locality and often renders the algorithm communication-bound.

*Hierarchical algorithms* exploit locality by organizing computation to respect data locality. They separate interactions into strong local components and weaker (or smoother) global components that can be efficiently compressed. The fundamental building block is the two-level decomposition. Let $n = \ell b$, where $\ell$ is the block size and $b$ is the number of blocks:

$$\boldsymbol{M} = \boldsymbol{D}_1 + \boldsymbol{U}_1 \boldsymbol{S}_1 \boldsymbol{V}_1^T$$

This structure separates the computation:

- $\boldsymbol{D}_1$ is a block-diagonal matrix. Computation is entirely local to input chunks, maximizing data reuse in fast memory (*near field*).

- $\boldsymbol{U}_1 \boldsymbol{S}_1 \boldsymbol{V}_1^T$ is a data-sparse (numerically low-rank) representation of global interactions between blocks (*far field*).

A fully hierarchical algorithm (e.g., FMM, $\mathcal{H}$-matrices) applies this concept recursively to the interaction matrix $\boldsymbol{S}_1$, creating a tree structure for computation.

| Feature | Flat Algorithms | Hierarchical Algorithms |
|---|---|---|
| Algebraic Structure | sequence of sparse factors | recursive low-rank block decomposition |
| Data Locality | low; global access patterns | high; maximizes local computation |
| Communication | global synchronization/shuffling | structured, hierarchical; reduced volume |
| Arithmetic Intensity | often lower; memory-bound | higher; high compute-to-memory ratio |

The input-to-state map of a scalar linear recurrence is defined by:

$$x_i = a_i x_{i-1} + u_i, \quad i \in [n] := \{1, \ldots, n\} \tag{1}$$

with *state* $x_i \in \mathbb{R}$, *input* $u_i \in \mathbb{R}$, and *coefficient* $a_i \in \mathbb{R}$, and initial condition $x_0 \in \mathbb{R}$. Our first analysis of such dynamics is purely algebraic and assumes exact arithmetic, holding for any sequence of coefficients without regard to numerical and analytical stability.

The global behavior of system (1) can be captured by the action of a linear operator $\boldsymbol{L} \in \mathbb{R}^{n \times n}$ on the input sequence. Let $x = (x_1, \ldots, x_n)$ and let $u = (u_1 + a_1 x_0, u_2, \ldots, u_n)$ so that the initial state is folded into the first input. We write $(p, q)$ for vertical concatenation, i.e., $(p, q) = [p^\top, q^\top]^\top$. Let $\boldsymbol{A} = \mathrm{diag}(a_1, \ldots, a_n)$ be the diagonal matrix of coefficients, and let $\boldsymbol{Z}$ be the down-shift operator ($\boldsymbol{Z}_{ij} = \delta_{i,j+1}$). The recurrence is equivalent to the vector equation $x = \boldsymbol{A}(\boldsymbol{Z}x) + u$, i.e. the state is the solution of the linear system:

$$(\boldsymbol{I} - \boldsymbol{A}\boldsymbol{Z})\,x = u. \tag{2}$$

The operator $\boldsymbol{I} - \boldsymbol{A}\boldsymbol{Z}$ is unit lower-triangular and therefore always invertible. The solution is $x = \boldsymbol{L}u$, where we define the system's *transfer operator* as $\boldsymbol{L} := (\boldsymbol{I} - \boldsymbol{A}\boldsymbol{Z})^{-1}$.

The structure of this operator is revealed through its Neumann series. The matrix $\boldsymbol{A}\boldsymbol{Z}$ is strictly lower-triangular and thus nilpotent, with $(\boldsymbol{A}\boldsymbol{Z})^n = 0$. Consequently, the series expansion for the inverse becomes a finite sum:

$$\boldsymbol{L} = \boldsymbol{I} + \boldsymbol{A}\boldsymbol{Z} + (\boldsymbol{A}\boldsymbol{Z})^2 + \cdots + (\boldsymbol{A}\boldsymbol{Z})^{n-1}. \tag{3}$$

Each term $(\boldsymbol{A}\boldsymbol{Z})^k$ in this expansion represents the propagation of influence forward by exactly $k$ steps. A direct computation (see Appendix A) shows that $(\boldsymbol{A}\boldsymbol{Z})^k$ is supported on its $k$-th sub-diagonal, with entries $\left[(\boldsymbol{A}\boldsymbol{Z})^k\right]_{i,i-k} = a_i a_{i-1} \cdots a_{i-k+1}$. To write this compactly, we define the product $a_{i:j} := a_i a_{i-1} \cdots a_j$ for

$$\boldsymbol{I} - \boldsymbol{A}\boldsymbol{Z} \qquad (\boldsymbol{I} - \boldsymbol{A}\boldsymbol{Z})^{-1}$$

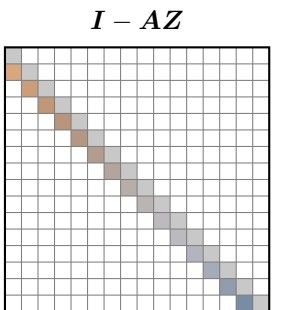 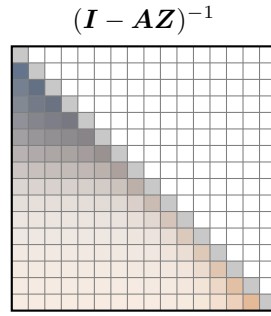

Figure 3.1: Side-by-side visualization of $\boldsymbol{I} - \boldsymbol{A}\boldsymbol{Z}$ and its Neumann-series inverse $(\boldsymbol{I} - \boldsymbol{A}\boldsymbol{Z})^{-1}$ for $n = 16$. Subdiagonals are colored by row index with opacity proportional to magnitude.

$i \geq j$ and adopt the convention that an empty product is 1 while if $i < j$ we have $a_{i:j} = 0$. The entries of $(\boldsymbol{A}\boldsymbol{Z})^k$ are thus $a_{i:i-k+1}$.

Summing the series yields the entries of the transfer operator. The entry $\boldsymbol{L}_{ij} = a_{i:j+1}$ captures the influence of input $u_j$ on state $x_i$, which is non-zero only if $i \geq j$. This gives the operator its characteristic unit lower-triangular form, depicted in Figure 3.1.

> *Sequentially Semi-Separable Matrices.* While dense, matrices of the type (3) possess a rich internal structure amenable to factorization, which is key to designing efficient algorithms. Commonly referred to as *sequentially semi-separable* matrices, they have been a long-time favorite of linear algebra and signal processing literature (in both finite and infinite dimensions). First introduced to study time-varying linear systems (Gohberg et al., 1992; Dewilde & Van der Veen, 1998), their properties have been extensively studied both theoretically (Vandebril et al., 2008; Dewilde & van der Veen, 2014; Dewilde et al., 2025) and computationally (Chandrasekaran et al., 2005). For practical-minded machine learning readers, Dao & Gu (2024) provide an excellent overview of the literature.

### 3.1 Matrix Factorizations and *Flat* Algorithms

A key contribution of our work is establishing a precise correspondence between classical parallel algorithms and sparse matrix factorizations—a connection that has been surprisingly underexplored despite its fundamental nature. We show that the transfer operator $\boldsymbol{L}$ admits a factorization into exactly $\log_2(n)$ sparse factors, directly mirroring the computational structure of parallel prefix scan algorithms. This insight transforms what appears to be a purely algebraic manipulation into a principled algorithmic framework.

The simplest factorization emerges naturally from the binary expansion of the geometric series (3), revealing that the Kogge-Stone parallel scan algorithm (Kogge & Stone, 2009) is not merely analogous to, but fundamentally *is*, a matrix factorization scheme. Assuming $n$ to be an integer power of two (enforceable by zero-padding $u$ if needed), $n = 2^m$ for some $m \in \mathbb{N}$, we have

$$\boldsymbol{L} \;=\; \sum_{k=0}^{n-1} (\boldsymbol{A}\boldsymbol{Z})^k \;=\; \prod_{t=0}^{m-1} \left( \boldsymbol{I} + (\boldsymbol{A}\boldsymbol{Z})^{2^t} \right), \tag{4}$$

where extra terms beyond $k = n - 1$ vanish by nilpotency[0]. Full derivation is provided in Appendix B. Figure 3.2 illustrates the sparsity structure of each factor in this decomposition.

This factorization immediately yields the Kogge-Stone parallel algorithm with logarithmic depth. Each factor $\left( \boldsymbol{I} + (\boldsymbol{A}\boldsymbol{Z})^{2^t} \right)$ in the product can be applied to the input vector $u$ through the recursive doubling scheme:

$$v \leftarrow v + \boldsymbol{F}v, \quad \boldsymbol{F} \leftarrow \boldsymbol{F}^2, \tag{5}$$

---

[0]The product sign applied to matrices is defined as the *left* matrix product, i.e., $\prod_{i=1}^{n} A_i = A_n \cdots A_2 A_1$.

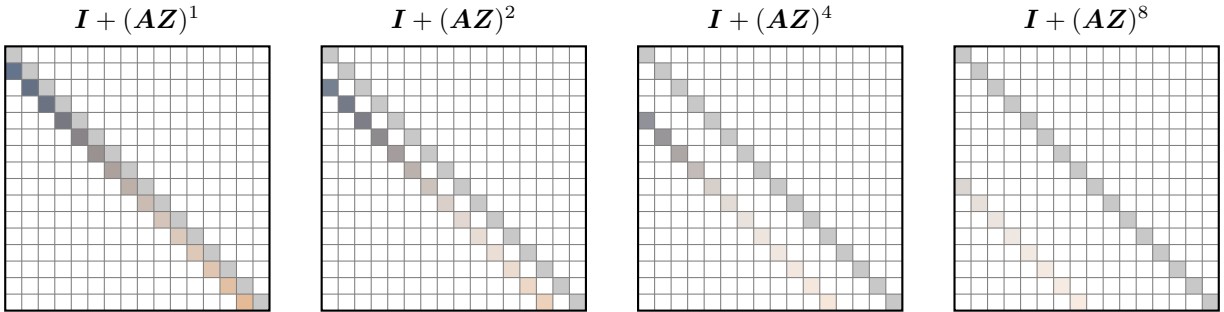

Figure 3.2: Kogge–Stone factorization of $\boldsymbol{L}$ for $n = 16$: the four sparse factors $\boldsymbol{I} + (\boldsymbol{AZ})^{2^t}$ for $t = \{0, 1, 2, 3\}$. Each matrix has ones on the main diagonal (gray) and a single subdiagonal at offset $2^t$ representing $(\boldsymbol{AZ})^{2^t}$.

initialized with $v = u$ and $\boldsymbol{F} = \boldsymbol{AZ}$. At stage $t$, we have $\boldsymbol{F} = (\boldsymbol{AZ})^{2^t}$. The efficiency of the algorithm stems the logarithmic depth combined with maintaining sparsity throughout: at each stage $t$, the matrix $\boldsymbol{F}$ remains a shifted diagonal supported on the $2^t$-th sub-diagonal, which we represent as $\boldsymbol{F} = \mathsf{diag}(f)\boldsymbol{Z}^{2^t}$. The squaring operation $\boldsymbol{F} \leftarrow \boldsymbol{F}^2$ can be computed implicitly through element-wise operations on the diagonal vector $f$, as detailed in the box below. This yields Algorithm 1, which implements the Kogge-Stone parallel prefix scan in our matrix formulation.

*Implicit matrix squaring.* Let $k = 2^t$. We define the *shift* operation on a vector $f$ such that $\mathsf{shift}(f, k)_i = f_{i-k}$ for $i > k$ and $0$ otherwise. The derivation relies on the following commutation identity between the shift operator and a diagonal matrix:

$$\boldsymbol{Z}^k \mathsf{diag}(f) = \mathsf{diag}(\mathsf{shift}(f, k))\boldsymbol{Z}^k.$$

Using this identity, the squaring step becomes:

$$\begin{aligned}
\boldsymbol{F}^2 &= \left(\mathsf{diag}(f)\boldsymbol{Z}^{2^t}\right)^2 \\
&= \mathsf{diag}(f)\left(\boldsymbol{Z}^{2^t}\mathsf{diag}(f)\right)\boldsymbol{Z}^{2^t} \\
&= \mathsf{diag}(f)\left(\mathsf{diag}(\mathsf{shift}(f, 2^t))\boldsymbol{Z}^{2^t}\right)\boldsymbol{Z}^{2^t} \\
&= \mathsf{diag}\left(f \odot \mathsf{shift}(f, 2^t)\right)\boldsymbol{Z}^{2^{t+1}},
\end{aligned}$$

where $\odot$ is the Hadamard product. This provides an efficient update for the vector $f$.

Algorithm 1 presents the Kogge-Stone variant, which achieves optimal $O(\log n)$ depth but performs $O(n \log n)$ total work. The matrix factorization perspective naturally extends to other parallel prefix algorithms: the Brent-Kung algorithm, for instance, corresponds to a different factorization that trades increased depth for work-efficiency, achieving $O(n)$ work with $O(\log n)$ depth through a two-phase (upsweep-downsweep) structure. We leave the proof as an exercise to the reader.

*Scan algorithms in deep learning.* Parallel scans have been widely used to parallelize linear recurrences over sequence length (Martin & Cundy, 2017; Smith et al., 2022; Gu & Dao, 2023). The typical recipe is: (1) define the associative binary operator $\circ : \mathbb{R}^2 \times \mathbb{R}^2 \to \mathbb{R}^2$ corresponding to the recurrence (Blelloch, 1990), $(v', f') \circ (v, f) := (v' + f'v, f'f)$; and (2) invoke a high-performance scan from standard libraries (e.g., `cub::DeviceScan`) (Merrill & Garland, 2016). On modern GPUs, optimized scans are memory-bandwidth limited and, for long sequences, reach throughput comparable to `memcpy` (Merrill & Garland, 2016; Harris et al., 2007), motivating algorithms that better match the hardware hierarchy.

---
**Algorithm 1** Kogge-Stone Parallel Algorithm

---
**Require:** $n \in \mathbb{N}$, sequences $(a_i)_{i=1}^n$, $(u_i)_{i=1}^n$ with $u_1 \leftarrow u_1 + a_1 x_0$ already folded
**Ensure:** $(x_i)_{i=1}^n$ where $x_i = \sum_{j=1}^i (a_{i:j}) \, u_j$

  1: $v \leftarrow (u_1, \ldots, u_n)^\top$                                                ▷ work vector
  2: $f \leftarrow (a_1, \ldots, a_n)^\top; \quad s \leftarrow 1$                            ▷ $\boldsymbol{F} = \mathrm{diag}(f) \, \boldsymbol{Z}^s$
  3: **while** $s \leq n - 1$ **do**                           ▷ stages $t = 0, \ldots, \lceil \log_2 n \rceil - 1$
  4:     **for all** $i \in \{s+1, \ldots, n\}$ **do**            ▷ apply $v \leftarrow v + Fv$ **in parallel**
  5:         $v_i \leftarrow v_i + f_i \cdot v_{i-s}$
  6:     **end for**
  7:     **for all** $i \in \{s+1, \ldots, n\}$ **do**        ▷ square $\boldsymbol{F}$: $f \leftarrow f \odot \mathrm{shift}(f, s)$ **in parallel**
  8:         $f_i \leftarrow f_i \cdot f_{i-s}$
  9:     **end for**
10:     $s \leftarrow 2s$
11: **end while**
12: **return** $x \leftarrow v$

---

Despite optimal $O(\log n)$ depth, flat factorizations impose global communication patterns that are bandwidth-bound on hierarchical hardware. We therefore reorganize the same operator around locality: partition the sequence into blocks, solve intra-block recurrences in parallel, and couple blocks only through a scalar carrier governed by a coarse recurrence, leading to the hierarchical formulation below.

## 3.2 Hierachical Decomposition and Algorithms

On *hierarchical* hardware — registers, caches, and memory arranged by increasing capacity and decreasing bandwidth — such sweeps squander locality. Effective algorithms design practice mirrors the memory hierarchy: partition the sequence into blocks and organize computation so that data remains near where it is produced.

We fix a block size $\ell \in \mathbb{N}$ and assume, for simplicity, that $n = b \cdot \ell$ for some $b \in \mathbb{N}$. We introduce a bijection $\phi : [b] \times [\ell] \to [n]$ mapping the block index $t$ and local index $j$ to the global index $i$ via $\phi(t, j) := \ell(t - 1) + j$ (essentially, row-major enumeration). This bijection allows us to segment any sequence $(q_1, \ldots, q_n)$. We denote the local components as $\boldsymbol{q}_{t,j} := q_{\phi(t,j)}$ and the block subvectors as $\boldsymbol{q}_t := (\boldsymbol{q}_{t,1}, \ldots, \boldsymbol{q}_{t,\ell})^\top \in \mathbb{R}^\ell$. We apply this notation specifically to the coefficients $a$, the input $u$, and the state $x$. We also adapt the notation for contiguous products to this block structure. Within a block $t$, we write $\boldsymbol{a}_{t,k:j} := \boldsymbol{a}_{t,k} \boldsymbol{a}_{t,k-1} \cdots \boldsymbol{a}_{t,j}$ for $k \geq j$, with the empty product (e.g., when $j = k + 1$) defined as unity.

### 3.2.1 Tiling the Transfer Operator

Under this partitioning, the transfer operator $\boldsymbol{L}$ naturally decomposes into a $b \times b$ block matrix, revealing the dependency structure between different blocks.

**Theorem 3.1** (Block structure). *The transfer operator $\boldsymbol{L}$ admits a block lower triangular representation*

$$
\boldsymbol{L} = \begin{bmatrix}
\boldsymbol{L}_1 & & & & \\
\boldsymbol{F}_{2,1} & \boldsymbol{L}_2 & & & \\
\boldsymbol{F}_{3,1} & \boldsymbol{F}_{3,2} & \boldsymbol{L}_3 & & \\
\vdots & \vdots & \ddots & \ddots & \\
\boldsymbol{F}_{b,1} & \boldsymbol{F}_{b,2} & \cdots & \boldsymbol{F}_{b,b-1} & \boldsymbol{L}_b
\end{bmatrix}, \tag{6}
$$

*where each diagonal block $\boldsymbol{L}_t \in \mathbb{R}^{\ell \times \ell}$ is unit lower triangular, and each off-diagonal block $\boldsymbol{F}_{t,s} \in \mathbb{R}^{\ell \times \ell}$ for $t > s$ has rank at most one.*

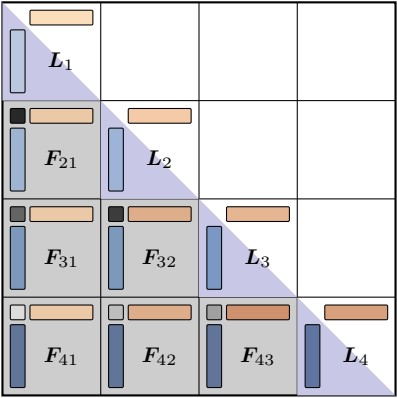

Figure 3.3: Block decomposition of the transfer operator showing diagonal blocks $\boldsymbol{L}_t = \mathrm{tril}(\boldsymbol{g}_t\boldsymbol{g}_t^{-\top})$ (lower triangular, capturing intra-block dependencies) and off-diagonal blocks $\boldsymbol{F}_{ts} = \boldsymbol{g}_t\beta_{ts}\boldsymbol{r}_s^\top$ (rank-one factorization, mediating inter-block carrier propagation). The scalar $\beta_{ts}$ represents the compound attenuation between blocks.

The global system $x = \boldsymbol{L}u$ thus decomposes into coupled equations for the state chunks:

$$\boldsymbol{x}_t = \boldsymbol{L}_t\boldsymbol{u}_t + \sum_{s=1}^{t-1} \boldsymbol{F}_{t,s}\boldsymbol{u}_s. \tag{7}$$

The term $\boldsymbol{L}_t\boldsymbol{u}_t$ represents the *intra-block* dynamics—the evolution of the state within block $t$ driven solely by the local inputs $\boldsymbol{u}_t$ and coefficients $\boldsymbol{a}_t$. The summation term captures the *inter-block* dynamics—the cumulative influence of all preceding blocks on the current block $t$. The intra-block terms $\boldsymbol{L}_t\boldsymbol{u}_t$ are independent and can be parallelized. The overall computational efficiency, however, hinges on effectively resolving the inter-block dependencies without explicitly computing materializing and multiplying the off-diagonal blocks of $\boldsymbol{L}$ (the summation in (7)).

**Intra-block dynamics.** The diagonal blocks $\boldsymbol{L}_t$ govern the intra-block dynamics. They characterize the evolution of the system within block $t$ assuming a zero initial state entering the block. When considering the relationship between an input and a state where both indices fall within the same block $t$, the transfer mechanism depends exclusively on the coefficients associated with that block.

Specifically, if we map global indices to local indices $(k, m)$, the entries are $[\boldsymbol{L}_t]_{k,m} = \boldsymbol{a}_{t,k:m+1}$ for $k > m$, and 1 for $k = m$. This structure is identical to the global transfer operator, but localized. If we define the local coefficient matrix $\boldsymbol{A}_t = \mathrm{diag}(\boldsymbol{a}_{t,1}, \ldots, \boldsymbol{a}_{t,\ell})$ and denote the $\ell \times \ell$ down-shift operator by $\boldsymbol{Z}_\ell$, the local transfer operator $\boldsymbol{L}_t$ can be compactly expressed as:

$$\boldsymbol{L}_t = (\boldsymbol{I}_\ell - \boldsymbol{A}_t\boldsymbol{Z}_\ell)^{-1}. \tag{8}$$

**Inter-block dynamics.** Information flowing from blocks 1 through $t-1$ into block $t$ must pass through a single scalar—the state $x_{\ell(t-1)}$ at the block boundary. This bottleneck, which we call the *carrier* $s_{t-1} := x_{\ell(t-1)} = \boldsymbol{x}_{t-1,\ell}$, compresses and mediates all inter-block dependencies and enables a hierarchical reformulation of inference algorithms.

Consider the state evolution within block $t$. For all $k \in [\ell]$, the state $\boldsymbol{x}_{t,k}$ can be expressed relative to the block entry point:

$$\boldsymbol{x}_{t,k} = \boldsymbol{a}_{t,k:1}s_{t-1} + \sum_{j=1}^{k} \boldsymbol{a}_{t,k:j+1}\boldsymbol{u}_{t,j}. \tag{9}$$

The first term propagates the incoming carrier through products $\boldsymbol{a}_{t,k:1}$, while the second captures local dynamics—precisely the $k$-th component of $\boldsymbol{L}_t\boldsymbol{u}_t$. Collecting these propagation factors into a vector $\boldsymbol{g}_t := (\boldsymbol{a}_{t,1}, \boldsymbol{a}_{t,2:1}, \ldots, \boldsymbol{a}_{t,\ell:1})$, we obtain the block state equation:

$$\boldsymbol{x}_t = \boldsymbol{L}_t\boldsymbol{u}_t + \boldsymbol{g}_ts_{t-1}. \tag{10}$$

The carrier itself evolves recurrently through the blocks —a *linear recurrence governing blocks of linear recurrences*. Extracting the final state $s_t = e_\ell^\top x_t$ and substituting (10):

$$s_t = c_t s_{t-1} + r_t^\top u_t, \tag{11}$$

where $c_t := a_{t,\ell:1}$ is the block's *compound attenuation* and $r_t^\top := e_\ell^\top L_t$ reads out the local contribution.

The carrier system inherits the algebraic structure of the original problem but operates at a coarser temporal resolution. This self-similarity allows us to apply the transfer operator formalism at this higher level as well.

Let $s = (s_1, \ldots, s_b)$ be the vector of carriers and $v = (v_1, \ldots, v_b)$ be the vector of effective inputs, $v_t = r_t^\top u_t$. The carrier dynamics can be written in matrix form as $s = CZ_b s + v$, where $C = \text{diag}(c_1, \ldots, c_b)$ is the diagonal matrix of block attenuations and $Z_b$ is the $b \times b$ down-shift operator.

Consequently, the carrier transfer operator $T \in \mathbb{R}^{b \times b}$, which resolves the inter-block dependencies via $s = Tv$, is given by

$$T = (I_b - CZ_b)^{-1}. \tag{12}$$

*Low-rank factorization of the off-diagonal blocks.* The constraint that the inter-block coupling must channel through the scalar carrier manifests as rank-one structure in the off-diagonal blocks.

**Theorem 3.2** (Low-rank Structure). *The off-diagonal blocks admit the factorization*

$$F_{t,s} = g_t \beta_{t,s} r_s^\top, \tag{13}$$

*where $\beta_{t,s} = c_{s+1} \cdots c_{t-1}$ compounds the block attenuations.*

*Proof.* The matrix and recurrence views must agree:

$$\sum_{s=1}^{t-1} F_{t,s} u_s = g_t s_{t-1}. \tag{14}$$

Solving the carrier recurrence from $s_0 = 0$ yields

$$s_{t-1} = \sum_{s=1}^{t-1} \beta_{t,s} v_s = \sum_{s=1}^{t-1} \beta_{t,s} (r_s^\top u_s). \tag{15}$$

Hence, we have

$$\sum_{s=1}^{t-1} F_{t,s} u_s = g_t \sum_{s=1}^{t-1} \beta_{t,s} (r_s^\top u_s)$$
$$= \sum_{s=1}^{t-1} (g_t \beta_{t,s} r_s^\top) u_s. \tag{16}$$

$\square$

The factorization $F_{t,s} = g_t \beta_{t,s} r_s^\top$ encodes the complete information flow: $r_s$ extracts the carrier from block $s$, $\beta_{t,s}$ attenuates it through intermediate blocks, and $g_t$ broadcasts it into block $t$.

These relations organize the computation into a two-level hierarchy: within each block, $x_{t,k} = a_{t,k} x_{t,k-1} + u_{t,k}$ with boundary condition $x_{t,1} = s_{t-1}$; across blocks, the carrier evolves as $s_t = a_{t,\ell:1} s_{t-1} + r_t^\top u_t$ with $s_1 = 0$.

Collecting the pieces, equations (10)–(11) show that inter-block influence is mediated entirely by the scalar carrier, and Theorem 3.2 expresses each off-diagonal tile as a rank-one map. Bundling these ingredients yields a hierarchical decomposition of the transfer operator (Figure 3.4). Introduce the block-diagonal aggregations $\mathcal{L} := \text{diag}(L_1, \ldots, L_b)$, $G := \text{diag}(g_1, \ldots, g_b)$, and $R := \text{diag}(r_1^\top, \ldots, r_b^\top)$, and let $T = (I_b - CZ_b)^{-1}$ be the carrier transfer operator.

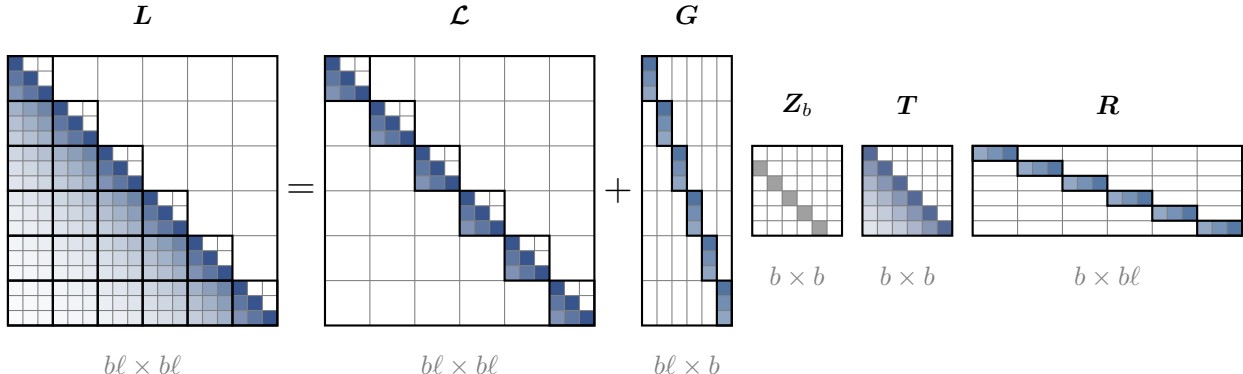

Figure 3.4: Transfer operator matrix $L$ with its hierarchical factorization $\mathcal{L} + GZ_bTR$. The full matrix exhibits exponentially decaying entries with darker blue diagonal blocks and lighter blue-gray subdiagonal blocks. The factorization separates local structure $\mathcal{L}$ (navy blue) from inter-block coupling via $G$ (blue-gray, decaying downward), $Z_b$ (gray shift), $T$ (slate blue-gray), and $R$ (blue-gray, decaying leftward).

**Theorem 3.3** (Hierarchical Decomposition). *With the notation above, the global transfer operator admits the exact decomposition*

$$L = \mathcal{L} \ + \ G\, Z_b\, T\, R. \tag{17}$$

*The first term collects independent intra-block solves; the second routes carriers across blocks.*

*Proof.* The $(t, s)$ block equals $L_t$ when $t = s$ (from $\mathcal{L}$). For $t > s$, the second term contributes $g_t[Z_bT]_{t,s}\, r_s^\top = g_t\, T_{t-1,s}\, r_s^\top$. Since $T = (I_b - CZ_b)^{-1}$ resolves the carrier recurrence (11), we have $T_{t-1,s} = \beta_{t,s}$, yielding $F_{t,s} = g_t\beta_{t,s}r_s^\top$. $\square$

### 3.2.2 From Factorization to Computation

The hierarchical factorization (17), translates directly into a structured, parallel algorithm for evaluating the linear recurrence (Algorithm 2). Each algebraic term maps precisely to a computational phase: the block-diagonal $\mathcal{L}$ corresponds to parallel local solves (Stage I); the carrier transfer operator $T$ drives global propagation (Stage II); and the rank-one factors $G$ and $R$ mediate the communication between these stages (Stage I extraction and Stage III reconstruction).

This structure is inherently aligned with modern memory hierarchies on AI accelerators. Local blocks are sized to saturate high-bandwidth cache or shared memory, while the compressed carrier system minimizes traffic across slower global memory interconnects.

Note that the quantities required for Stage II and III are readily available from the local computation in Stage I. The effective input $v_t$ is the final component of the local state $w_t$, $v_t = w_{t,\ell}$. Further, if $L_t$ is materialized, $g_t$ corresponds to its first column scaled by $a_{t,1}$, $g_t = a_{t,1}L_te_1$, and $c_t = g_{t,\ell}$.

**Implementation strategies.** The abstract recurrences in Algorithm 2 (lines 2 and 7) can be implemented using different strategies optimized for specific hardware characteristics. While alternatives include sequential scans (work-optimal but serial) and parallel scans (logarithmic depth but higher work; see Section 3.1), we focus on the strategy best suited for modern accelerators: matrix multiplication.

This approach materializes the transfer operators ($L_t$ locally, $T$ globally) as dense matrices, transforming the recurrence solve into a GEMM. While this increases the local work complexity to $O(\ell^2)$, compared to the $O(\ell)$ work of a sequential scan, it maximizes arithmetic intensity.

This strategy is particularly potent when the input $u$ has a feature dimension $d$ (or other parallel dimensions such as batch size) where the coefficients $a$ are shared. The operation becomes a matrix-matrix multiplication, and the $O(\ell^2)$ cost of materializing $L_t$ is amortized across the parallel dimensions. This formulation

---

**Algorithm 2** HIERARCHICAL BLOCK PARALLEL INFERENCE

---

**Require:** Block structure $n = b\ell$, inputs $(\boldsymbol{u}_t)_{t=1}^b$, coefficients $(\boldsymbol{a}_t)_{t=1}^b$
**Ensure:** State sequence $(\boldsymbol{x}_t)_{t=1}^b$

    STAGE I: LOCAL SOLVES AND INTERFACE EXTRACTION

1: **for all** $t \in [b]$ **in parallel do**
2:      $\boldsymbol{w}_t \leftarrow \boldsymbol{L}_t \boldsymbol{u}_t$          $\triangleright$ Solve local recurrence
3:      $\boldsymbol{g}_t \leftarrow [\boldsymbol{a}_{t,1}, \boldsymbol{a}_{t,2:1}, \ldots, \boldsymbol{a}_{t,\ell:1}]^\top$          $\triangleright$ Propagation factors
4:      $v_t \leftarrow \boldsymbol{w}_{t,\ell}; \quad c_t \leftarrow \boldsymbol{g}_{t,\ell}$          $\triangleright$ Extract interface
5: **end for**
    STAGE II: GLOBAL CARRIER RECURRENCE
6: Form $\boldsymbol{C} = \mathrm{diag}(c_1, \ldots, c_b)$ and $\boldsymbol{v} = [v_1, \ldots, v_b]^\top$
7: $\boldsymbol{s} \leftarrow (\boldsymbol{I}_b - \boldsymbol{C}\boldsymbol{Z}_b)^{-1}\boldsymbol{v}$          $\triangleright$ Solve carrier system
    STAGE III: RECONSTRUCTION
8: $s_0 \leftarrow 0$
9: **for all** $t \in [b]$ **in parallel do**
10:      $\boldsymbol{x}_t \leftarrow \boldsymbol{w}_t + \boldsymbol{g}_t s_{t-1}$          $\triangleright$ Combine local and global
11: **end for**

---

leverages specialized hardware (e.g., tensor cores), often achieving peak utilization where scan-based methods remain bandwidth-limited.

While matrix multiplication is typically optimal locally, if the number of blocks $b$ is exceedingly large, the $O(b^2)$ cost of the global stage may become prohibitive. In such cases, a hybrid approach is viable: using matrix multiplication locally, and a parallel scan globally to resolve the carrier system in $O(b \log b)$ work and $O(\log b)$ depth.

*Numerically stable materialization of semi-separable matrices.* The matrix multiplication strategy requires materializing transfer operators whose entries $\boldsymbol{L}_{t,ij} = \boldsymbol{a}_{t,i:j+1}$ are products of at most $\ell - 1$ coefficients. One idea is to exploit the identity $\boldsymbol{a}_{t,i:j+1} = \boldsymbol{a}_{t,i:1}/\boldsymbol{a}_{t,j:1}$ for $i > j$ (Vandebril et al., 2008). Equivalently,

$$\boldsymbol{L}_t = \mathrm{tril}(\boldsymbol{g}_t \boldsymbol{g}_t^{-\top}) \tag{18}$$

where $\boldsymbol{g}_t^{-1}$ is intended to denote the reciprocal of $\boldsymbol{g}_t$, $\boldsymbol{g}_t^{-1} = (1/a_{t,1}, 1/a_{t,2:1}, \ldots, 1/a_{t,\ell:1})$. Naively forming the cumulative products $\boldsymbol{g}_t$ and compute their outer ratio is numerically fragile: when coefficients are contractive, $\boldsymbol{g}_{t,i}$ rapidly underflows to zero in low precision formats (e.g., fp16/bf16) producing indeterminate forms (0/0) even if the ratio $\boldsymbol{g}_{t,i}/\boldsymbol{g}_{t,j}$ is representable.
A standard remedy is to work with log-prefix sums $\boldsymbol{p}_{t,i} = \sum_{k=1}^i \log \boldsymbol{a}_{t,k}$ and set

$$\log \boldsymbol{L}_{t,ij} = \boldsymbol{p}_{t,i} - \boldsymbol{p}_{t,j} \quad \text{for } i \geq j, \tag{19}$$

i.e., form the outer difference $\log \boldsymbol{L}_t = \boldsymbol{p}_t \mathbf{1}^\top - \mathbf{1}\boldsymbol{p}_t^\top$ and mask the upper triangle (set to $-\infty$). This avoids explicit $\boldsymbol{g}_t$ and only exponentiates differences. However, subtracting large, nearly equal prefixes near the diagonal can introduce cancellation. Dao (2024) replaces the outer difference by a masked cumulative *sum* of the segment itself: tile $\log \boldsymbol{a}_t$ into a matrix $(\log \boldsymbol{a}_t) \otimes \mathbf{1}$ (repeating each segment $\ell$ times), zero out the inclusive upper triangle, perform a column-wise cumulative sum, then mask the strict upper triangle to $-\infty$ and exponentiate, avoiding subtractive cancellation. Dao's log-space route is particularly natural when the model parameterizes $\boldsymbol{a}_{t,i} = \exp(\gamma_{t,i})$ with $\gamma_{t,i} \leq 0$ (no extra log is needed).
In our setting we typically parameterize $\boldsymbol{a}_{t,i} = \sigma(u_{t,i}) \in (0,1)$ with a sigmoid (see Section 5); we do not get the first log "for free." We therefore introduce a *linear-space analogue* of Dao's construction that avoids subtraction and avoids unnecessary log / exp transforms. Expand $\boldsymbol{a}_t$ as $\boldsymbol{a}_t \otimes \mathbf{1}$, set the inclusive upper triangle to the multiplicative identity 1, take a column-wise cumulative *product*, and finally zero the strict upper triangle.

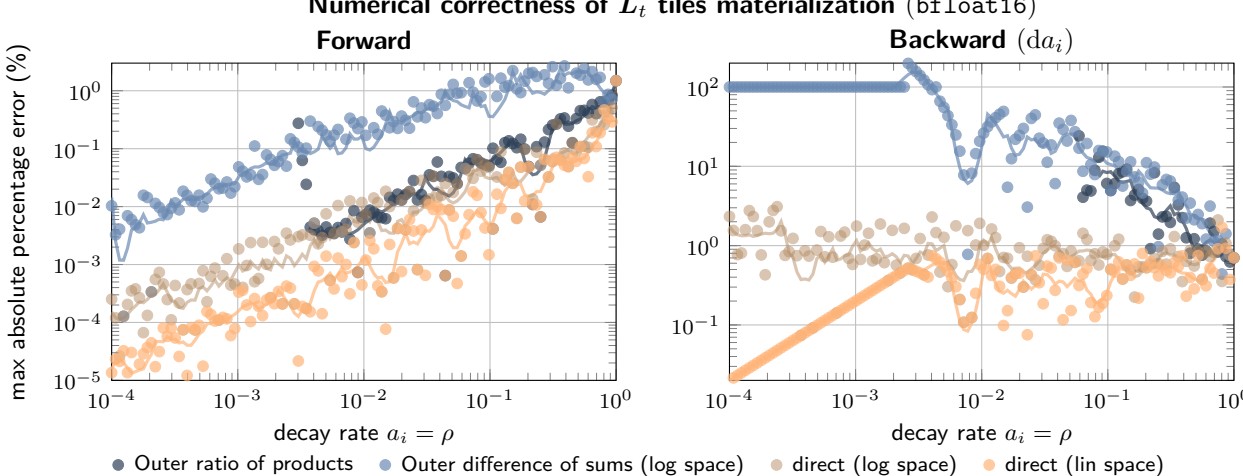

Figure 3.5: Numerical correctness of $\boldsymbol{L}_t$ tiles materialization. For each decay rate $\rho \in [10^{-4}, 1]$, we construct a block of size $\ell = 16$ with constant decay values $a_i = \rho$. Each method computes the $\boldsymbol{L}$ matrix in `bfloat16` and is compared against a `float64` reference implementation. We report the maximum absolute error normalized by the maximum magnitude (percentage error) for both the forward pass (left) and backward pass gradient with respect to $a_i$ (right).

This achieves the materialization robustly with complexity $O(b\ell^2)$. Zeros in $\boldsymbol{a}_t$ propagate correctly (downstream products become exactly 0). In practice, one could further accumulate in fp32 to extend dynamic range; empirically, we find the linear-space variant to match the intended semantics more closely and to be robust at the block sizes we use (see Figure 3.5). Note that the same construction applies to the global carrier $\boldsymbol{T}$. In optimized CUDA implementations, this algorithm is executed by a single warp, spawning one warp per chunk $t$ of the sequence.

---

Algorithm 3: DIRECT (LINEAR-SPACE) MATERIALIZATION OF TRANSFER OPERATOR

---

**Require:** Coefficients $\boldsymbol{a}_t \in \mathbb{R}^\ell$
**Ensure:** Transfer operator $\boldsymbol{L}_t \in \mathbb{R}^{\ell \times \ell}$
1: $A \leftarrow \boldsymbol{a}_t \mathbf{1}^\top$             ▷ Tile along columns: $A_{ij} = a_{t,i}$
2: $A_{ij} \leftarrow 1$ for $i \leq j$         ▷ Pre-mask with multiplicative identity
3: $P \leftarrow \mathsf{CumProd}_\downarrow(A)$         ▷ Column-wise cumulative product
4: $\boldsymbol{L}_t \leftarrow \mathrm{tril}(P)$       ▷ Causal lower triangle; strict upper set to 0

---

**Complexity and hardware utilization.** We focus on the primary strategy utilizing matrix multiplication for both stages, assuming a parallel feature dimension $d$. The total computational cost is dominated by the matrix multiplications. Locally, the cost is $O(b(\ell^2 d + \ell^2))$, where $b\ell^2 d$ is the cost of the multiplications and $b\ell^2$ is the cost of materialization across all $b$ blocks. Globally, the cost is $O(b^2 d + b^2)$.

While the total FLOP count exceeds the $O(nd)$ cost of a sequential scan by a factor related to $\ell$ and $b$, the arithmetic intensity compensates significantly. For the local stage, the arithmetic intensity scales favorably with both $\ell$ and $d$. This high intensity is the key to achieving high throughput on modern GPUs, enabling effective utilization of specialized matrix-multiplication hardware. If the global cost dominates due to a very large $b$, utilizing a parallel scan globally reduces the global work to $O(bd \log b)$.

*Hierarchical algorithms in practice.* Dao (2024) implements local solves as matmul kernels that can target Tensor Cores via `wmma` on SM90/SM100, using mixed precision (e.g., fp16/bf16 with fp32 accumulation) when heads share coefficients across features; it transitions from matmul to scan for

the global carrier recurrence above a certain number of blocks. By contrast, other designs such as Yang et al. (2023) or Yang et al. (2024) often favor scan-based local aggregation in full precision, paired with short-range or sequential carrier propagation; this trades reduced global communication for extra depth and does not make use of Tensor Cores. Note that even `cub::DeviceScan` is hierarchical (Merrill & Garland, 2016), so practical kernels de facto nest multiple hierarchies. As sequences grow, the carrier recurrence over $b$ blocks can itself become a bottleneck (motivating parallel scans globally, or further nesting). Because the carrier system possesses the same algebraic structure as the original recurrence, it can be recursively block-partitioned and factorized. This multi-level nesting aligns the computation with deeper hardware hierarchies (e.g., thread blocks, SMs, devices), preserving the rank-one structure between levels.

While hierarchical algorithms successfully reduce communication volume compared to flat scans and maintain logarithmic or even constant depth—avoiding the $O(n)$ depth of sequential global propagation—they cannot eliminate global synchronization entirely. The carrier recurrence, though compressed, still requires communication across all $b$ blocks. Moreover, this global coupling becomes numerically delicate when $c_t = \boldsymbol{a}_{t,\ell:1}$ approaches machine precision for large block counts.

To achieve truly local computation, we must sacrifice the global range of the recurrence. This is not a universal claim about all recurrence regimes: when coefficients remain near one, the recurrence approaches a global accumulation mechanism. Rather, SWR targets the contractive regime, where products along recurrent paths decay with range and appear as diminishing entries along the subdiagonals of the transfer operator. This natural decay suggests that long-range dependencies can become negligible, making truncation a useful efficiency trade-off in the regimes we study.

A naive truncation to purely independent blocks would sever all inter-block dependencies, yielding unacceptable accuracy. Instead, we pursue a principled middle ground: retaining nearest-neighbor communication between adjacent blocks while eliminating global synchronization. This is accomplished by early-stopping the carrier system after its first term—equivalent to approximating $\boldsymbol{T} \approx \boldsymbol{I}_b$ in our matrix formulation. The resulting operator preserves the first block-subdiagonal, achieving constant $O(1)$ depth, linear $O(n)$ work, and exclusively local communication patterns. This leads to the *Sliding Window Recurrence* variants developed in the next section.

# 4   Sliding Window Recurrences

The hierarchical algorithms of Section 3.2 achieve substantial reductions in communication volume while maintaining logarithmic depth. Yet, they do not eliminate the persistent trade-off between algorithmic depth and parallelizability versus communication efficiency: the scalar carrier recurrence still necessitates global synchronization across all $b$ blocks, emerging as a scalability bottleneck for increasingly long sequences.

These challenges compel us to reconsider the fundamental trade-off between global context and efficiency. In stable linear systems, the only ones trained in practice, where $|a_i| \leq \rho < 1$, the transfer operator exhibits a remarkable property: its entries decay exponentially along subdiagonals. Specifically, the influence of input $u_j$ on state $x_i$ is bounded by $\rho^{i-j}$, vanishing rapidly with temporal lag $i - j$. This structure suggests that enforcing global dependencies may be unnecessarily conservative since distant inputs contribute negligibly to local state evolution.

This insight motivates *Sliding Window Recurrences* (SWRs), a family of algorithms that strategically truncate the computational horizon to achieve local, embarrassingly parallel computation. Rather than computing the full dense transfer operator $\boldsymbol{L}$, we construct structured approximations that preserve essential dependencies while discarding those below numerical significance. We introduce two complementary truncation strategies:

> i. **Uniform Window.** Early termination of the flat parallel scan algorithm 1 yields a banded transfer operator capturing dependencies up to a fixed lag $k$. While theoretically appealing with $O(n \log k)$ work and $O(\log k)$ depth, this method inherits the communication-bound characteristics of its par-

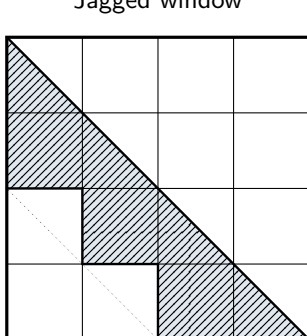

Figure 4.1: Comparison of the sparsity pattern of uniform and jagged sliding window recurrences. Hatched regions indicate the included support.

ent algorithm. On hierarchical hardware, it offers limited practical advantage over hierarchical appraches, serving primarily as a theoretical baseline.

ii. **Jagged Window.** Strategic truncation of the hierarchical carrier system, specifically the approximation $T \approx I_b$, yields a block-bidiagonal structure that preserves all intra-block and adjacent-block couplings. The resulting Block Two-Pass (B2P) algorithm eliminates global synchronization entirely, achieving constant $O(1)$ depth with purely local communication. This approach maintains the architectural advantages of hierarchical decomposition while enabling massive parallelism.

Our development prioritizes the jagged window variant, which aligns naturally with the memory hierarchies and execution models of modern accelerators. The uniform window, while included for completeness, serves primarily to establish theoretical context and performance bounds. The remainder of this section formalizes the computational horizon, presents both truncation strategies with their algorithmic realizations, and quantifies the accuracy-efficiency trade-offs through rigorous error analysis.

*Computational Horizon.* In contractive systems where $|a_i| \leq \rho < 1$ for some contraction rate $\rho$, the dependencies in the transfer operator $L$ decay exponentially with distance. This decay establishes a computational horizon: a finite effective range beyond which distant inputs have negligible influence on the current state. By exploiting this structure through early stopping in parallel algorithms, we can achieve significant computational efficiencies while maintaining controlled accuracy.

The influence of input $u_j$ on state $x_i$ (for $i > j$) is given by the product $a_{i:j}$, whose magnitude is bounded by $|a_{i:j}| \leq \rho^{i-j}$. As the lag $\ell = i - j$ increases, this influence diminishes exponentially. To quantify the computational horizon, consider a desired accuracy level $\varepsilon > 0$. A pointwise bound requires the minimal $k_\varepsilon$ such that $\rho^\ell < \varepsilon$ for all $\ell > k_\varepsilon$, which yields

$$k_\varepsilon = \left\lceil \frac{\log \varepsilon}{\log \rho} \right\rceil, \tag{20}$$

since $\log \rho < 0$ ensures subsequent terms remain smaller. However, for a more precise control over the cumulative error in the state, we consider the tail sum bound: the effective bandwidth $k$ is the smallest integer satisfying

$$\sum_{\ell=k+1}^{\infty} \rho^\ell = \frac{\rho^{k+1}}{1 - \rho} < \varepsilon. \tag{21}$$

Solving gives $k = \lceil \log(\varepsilon(1 - \rho))/\log \rho \rceil - 1$. This tail bound, assuming bounded inputs $|u_i| \leq \nu$, ensures the total contribution from distant inputs is below $\varepsilon\nu$.

Notably, finite precision imposes an absolute ceiling on the effective range, determined by the largest lag before the product of coefficients underflows to zero. The slowest possible decay, corresponding

to the largest contraction factor $\rho < 1$ representable in a given format, sets this upper bound. This maximum horizon is therefore an intrinsic property of the number system itself.

| Format | $p$ | $e$ bits | Bias | $\rho = 1 - 2^{-p-1}$ | $\epsilon$ | $\epsilon \cdot 2^{-p}$ | $k$ (normal) | $k$ (with subnormals) |
|---|---|---|---|---|---|---|---|---|
| fp32 | 23 | 8 | 127 | 0.999999940395... | $2^{-126}$ | $2^{-149}$ | 1465264032 | 1732732863 |
| fp16 | 10 | 5 | 15 | 0.999511718750 | $2^{-14}$ | $2^{-24}$ | 19869 | 34061 |
| bf16 | 7 | 8 | 127 | 0.996093750000 | $2^{-126}$ | $2^{-133}$ | 22314 | 23554 |
| fp8 e5m2 | 2 | 5 | 15 | 0.875000000000 | $2^{-14}$ | $2^{-16}$ | 72 | 83 |
| fp8 e4m3 | 3 | 4 | 7 | 0.937500000000 | $2^{-6}$ | $2^{-9}$ | 64 | 96 |

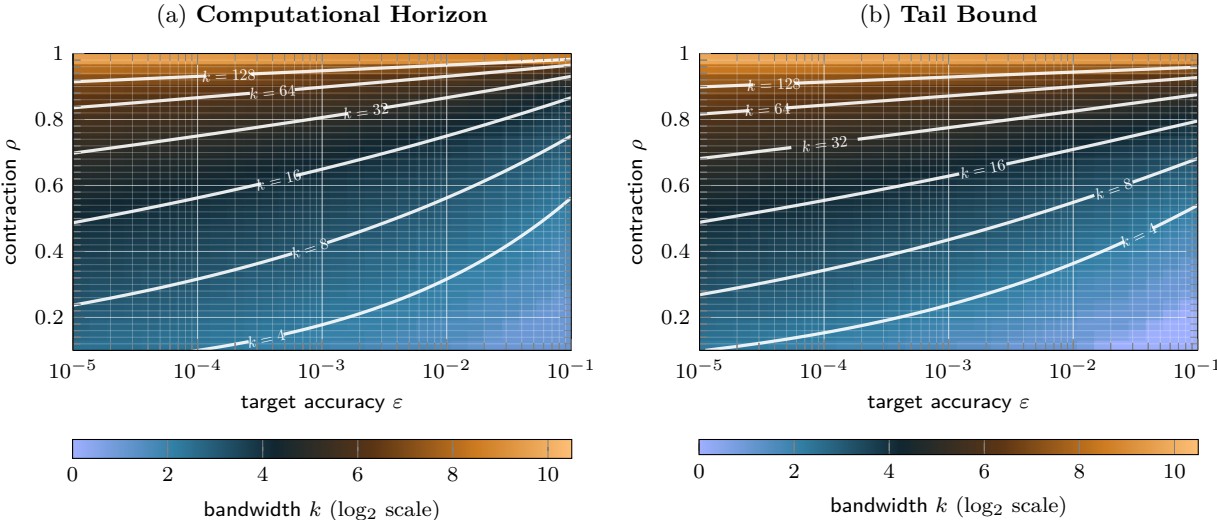

(a) **Computational Horizon**  (b) **Tail Bound**

Figure 4.2: Bandwidth requirements for finite-range approximations. (a) Computational horizon: required bandwidth $k$ to achieve target accuracy $\varepsilon$ for a contraction $\rho$. (b) Tail bound: required bandwidth $k$ so that the tail sum $\sum_{\ell=k+1}^{\infty} \rho^\ell < \varepsilon$. Heat maps show $\log_2 k$; white contours mark power-of-2 bandwidth values.

### 4.1 Uniform Window Recurrences

The uniform window approach truncates the Kogge-Stone factorization at a predetermined stage, yielding a banded approximation of the transfer operator. While conceptually straightforward, this strategy inherits the communication patterns of flat algorithms, limiting its practical utility on hierarchical hardware. We present it briefly for theoretical completeness.

At stage $\log_2 k$ of the Kogge-Stone algorithm, coefficient products have decayed to magnitude below $\rho^k$. When this falls below threshold $\varepsilon$, subsequent stages contribute negligibly. Terminating after $\log_2 k$ stages yields the truncated operator:

$$\tilde{L} = I + AZ + \cdots + (AZ)^{k-1} \tag{22}$$

This captures dependencies up to lag $k$, achieving work complexity $O(n \log k)$ and depth $O(\log k)$. For fixed bandwidth $k$, this reduces to linear work with constant depth. Figure 4.3 illustrates this banded structure.

### 4.2 Jagged Window Recurrences and the Block Two-Pass Algorithm

The hierarchical decomposition of Section 3.2 provides a principled framework for locality-preserving truncation. We exploit the algebraic structure of the carrier system to achieve a careful balance: preserving essential local and near-neighbor interactions while eliminating costly global synchronization.

**Truncation of the carrier system**  The exact hierarchical factorization (17) expresses the transfer operator as $L = \mathcal{L} + GZ_bTR$, where inter-block dependencies flow through the carrier transfer operator

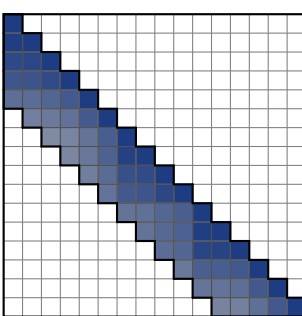

Figure 4.3: **Uniform window truncation.** The full transfer operator $\boldsymbol{L}$ (left) and its banded approximation $\tilde{\boldsymbol{L}}$ (right) from Eq. (22), capturing dependencies up to lag $k = 5$. Early termination of the Kogge-Stone factorization yields $O(n \log k)$ work complexity.

$\boldsymbol{T} = (\boldsymbol{I}_b - \boldsymbol{C}\boldsymbol{Z}_b)^{-1}$. Expanding the Neumann series we obtain:

$$\boldsymbol{T} = \boldsymbol{I}_b + \boldsymbol{C}\boldsymbol{Z}_b + (\boldsymbol{C}\boldsymbol{Z}_b)^2 + \cdots + (\boldsymbol{C}\boldsymbol{Z}_b)^{b-1}. \tag{23}$$

Each term represents increasingly distant inter-block interactions, attenuated by products of block coefficients. In stable systems, these products decay exponentially and are likely to be negligible: these are partial products of the coefficients $c_t = \boldsymbol{a}_{t,\ell:1}$ which are already bounded by $\rho^\ell$. This suggests a natural truncation point. We adopt the most aggressive meaningful approximation: retaining only the identity term, $\boldsymbol{T} \approx \boldsymbol{I}_b$. This preserves direct influence between adjacent blocks while discarding the accumulative carrier dynamics. It yields the *jagged* window transfer operator:

$$\tilde{\boldsymbol{L}} = \mathcal{L} + \boldsymbol{G}\boldsymbol{Z}_b\boldsymbol{R}. \tag{24}$$

The resulting structure preserves exact dynamics within blocks and the coupling between neighbors, capturing the essential local evolution while sacrificing only the rapidly-decaying global correlations. In practice, captures all dependencies up to lag $\ell$ while the *jagged* part captures part of the dependencies up to lag $2\ell - 1$ (Less error than uniform truncation with $k = \ell$ but more error than uniform truncation with $k = 2\ell - 1$). There is therefore no error on the first $2\ell$ time steps. As illustrated in Figure 4.4, $\tilde{\boldsymbol{L}}$ exhibits a distinctive block-bidiagonal structure that we term a "*jagged*" window:

$$\tilde{\boldsymbol{L}} = \begin{bmatrix} \boldsymbol{L}_1 & & & \\ \boldsymbol{F}_{2,1} & \boldsymbol{L}_2 & & \\ & \ddots & \ddots & \\ & & \boldsymbol{F}_{b,b-1} & \boldsymbol{L}_b \end{bmatrix}, \quad \boldsymbol{F}_{t,t-1} = \boldsymbol{g}_t \boldsymbol{r}_{t-1}^\top \tag{25}$$

The block structure is visualized in Figure 4.5, showing the diagonal blocks $\boldsymbol{L}_t$ and the rank-one off-diagonal blocks $\boldsymbol{F}_{t,t-1}$. The corresponding input-to-state mapping simplifies remarkably:

$$\tilde{\boldsymbol{x}}_t = \boldsymbol{L}_t \boldsymbol{u}_t + \boldsymbol{F}_{t,t-1} \boldsymbol{u}_{t-1}. \tag{26}$$

Each block's state depends only on its own inputs and those of its immediate predecessor.

### 4.2.1 The Block Two-Pass Algorithm

The truncation $\boldsymbol{T} \approx \boldsymbol{I}_b$ transforms the three-stage hierarchical algorithm into an elegant two-pass procedure where the global carrier solve vanishes entirely, replaced by direct propagation of local effective inputs between adjacent blocks. The first pass utilizes matrix multiplications for the local recurrences $\boldsymbol{w}_t = \boldsymbol{L}_t \boldsymbol{u}_t$, performing independent local solves across all blocks simultaneously while computing both the local state evolution and extracting the boundary value that serves as an approximate carrier, which is simply the last

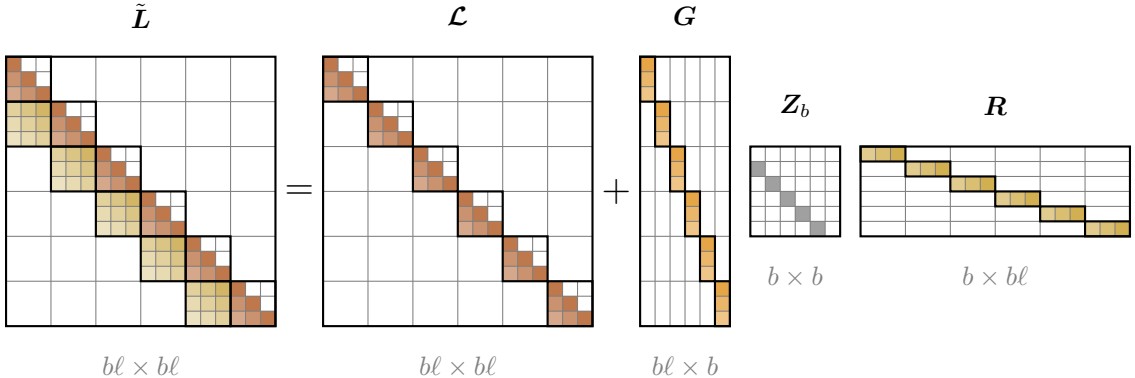

Figure 4.4: Jagged window approximation $\tilde{\boldsymbol{L}} = \boldsymbol{\mathcal{L}} + \boldsymbol{G}\boldsymbol{Z}_b\boldsymbol{R}$. The factorization separates local structure $\boldsymbol{\mathcal{L}}$ (medium orange) from inter-block coupling via $\boldsymbol{G}$, $\boldsymbol{Z}_b$, and $\boldsymbol{R}$.

component $s_t = v_t = \boldsymbol{w}_{t,\ell}$. The second pass reconstructs the full state as $\tilde{\boldsymbol{x}}_t = \boldsymbol{w}_t + \boldsymbol{g}_t v_{t-1}$ by injecting these boundary values into adjacent blocks. This B2P algorithm achieves optimal complexity for local parallel computation with constant depth $O(1)$ and work $O(b\ell^2 d + b\ell d) = O(nd)$. Making it linear in sequence length for fixed block size $\ell$.

---

**Algorithm 4** BLOCK TWO-PASS ALGORITHM (B2P)

---

**Require:** Block structure $n = b\ell$, inputs $(\boldsymbol{u}_t)_{t=1}^b$, coefficients $(\boldsymbol{a}_t)_{t=1}^b$
**Ensure:** Approximate state sequence $(\hat{\boldsymbol{x}}_t)_{t=1}^b$
    PASS I: PARALLEL LOCAL SOLVES
1: **for all** $t \in [b]$ **in parallel do**
2:     Materialize $\boldsymbol{L}_t$ and $\boldsymbol{g}_t$
3:     $\boldsymbol{w}_t \leftarrow \boldsymbol{L}_t \boldsymbol{u}_t$                                           $\triangleright$ Local solve via GEMM
4:     $v_t \leftarrow \boldsymbol{w}_{t,\ell}$                                            $\triangleright$ Extract effective input
5: **end for**
    PASS II: PARALLEL RECONSTRUCTION WITH SHIFT
6: $v_0 \leftarrow 0$
7: **for all** $t \in [b]$ **in parallel do**
8:     $\hat{\boldsymbol{x}}_t \leftarrow \boldsymbol{w}_t + \boldsymbol{g}_t v_{t-1}$                                  $\triangleright$ Inject neighbor contribution
9: **end for**

---

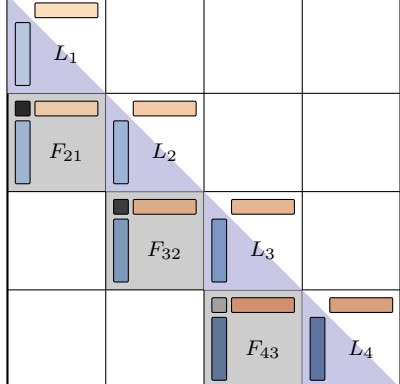

Figure 4.5: Block decomposition of the transfer operator showing diagonal blocks $\boldsymbol{L}_t = \mathrm{tril}(\boldsymbol{g}_t \boldsymbol{g}_t^{-\top})$ (lower triangular, capturing intra-block dependencies) and off-diagonal blocks $\boldsymbol{F}_{ts} = \boldsymbol{g}_t \beta_{ts} \boldsymbol{r}_s^\top$ (rank-one factorization, mediating interblock carrier propagation). The scalar $\beta_{ts}$ represents the compound attenuation between blocks.

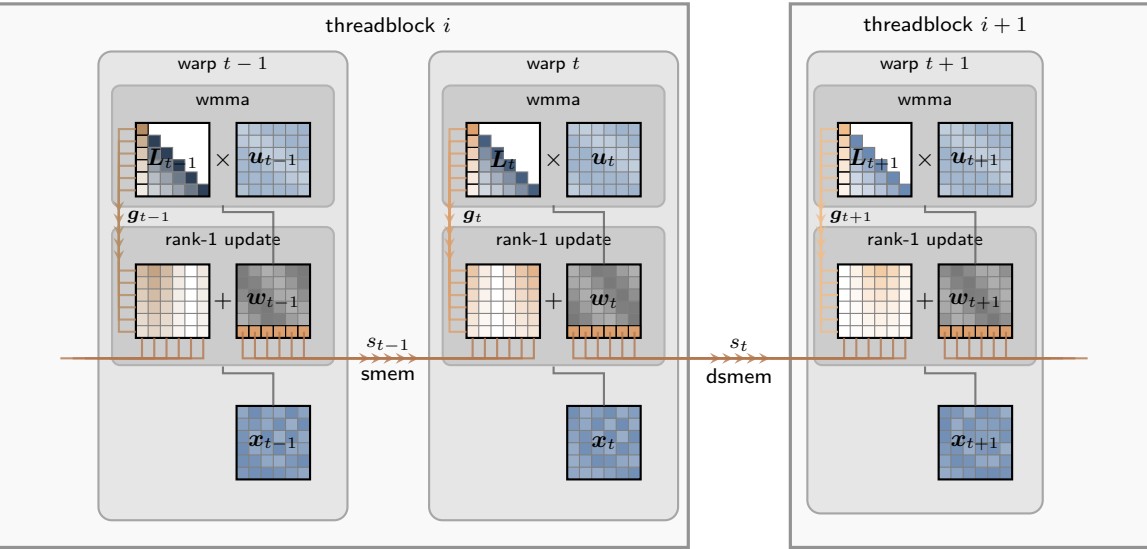

Figure 4.6: Block Two-Pass across three consecutive blocks. Top: per-block WMMA matmul $\boldsymbol{L}_t\boldsymbol{u}_t$ with head dimension $\ell$. Bottom: each pair runs in a single GPU warp; the carrier $s_t$ is sent to the next warp for the rank-1 reconstruction. Intra-threadblock communication uses shared memory (smem) while inter-threadblock communication uses distributed shared memory (dsmem).

### 4.2.2 Hardware Realization on Modern GPUs

The B2P algorithm is designed to map directly onto the memory and execution hierarchies of modern accelerators. We tailor its implementation for NVIDIA GPUs by aligning the block size $\ell$ with the dimensions of warp-level matrix multiply-accumulate (`wmma`) instructions, which target Tensor Cores and operate on small, fixed-size tiles. By setting the block size to $\ell = 16$, we can assign the computation for each time block to a single 32-thread warp, maximizing hardware utilization. A cooperative thread array (CTA, or thread block), a group of warps running on a single streaming multiprocessor (SM), can then process a larger sequence segment, communicating intermediate results via fast on-chip shared memory (SMEM). On recent architectures, clusters of CTAs can further co-operate using distributed shared memory (DSMEM) for low-latency exchange across a larger portion of the chip.

The parametrization of the recurrence (Section 5) is directly guided by the underlying hardware. Tensor Cores are optimized for dense matrix-matrix multiplications. A naive implementation that processes each of the $d$ feature channels independently with a matrix-vector product would fail to leverage these cores. To maximize hardware utilization and following previous work (Dao, 2024; Ku et al., 2025), we structure the computation into heads. Within each head, the same set of recurrence coefficients is shared across all $d$ feature dimensions. This architectural choice allows us to treat an entire input block $\boldsymbol{u}_t \in \mathbb{R}^{\ell \times d}$ as a dense matrix. The local solve $\boldsymbol{w}_t = \boldsymbol{L}_t\boldsymbol{u}_t$ is then computed as a single matrix-matrix product, matching the execution model of Tensor Cores. We typically set the feature dimension per head to $d = 16$, aligning with the $16 \times 16$ tile size of `wmma` instructions.

**Design choices.** The block size $\ell = 16$ is therefore not an isolated modeling hyperparameter: it fixes the tensor-core tile shape and the B2P kernel schedule. It is also the most aggressive practical truncation supported by this implementation. Choosing the smallest efficient tile exposes the speed-quality trade-off directly. A meaningful sweep over $\ell$ would require separately optimized kernels rather than a configuration change alone. A uniform band at comparable radius leaves many tile entries masked, underutilizing the GPU; the jagged window keeps local tiles dense and adds only the rank-one nearest-neighbor tile. Finally, sharing recurrence coefficients within a head is what turns $d$ independent matrix-vector recurrences into one $\ell \times d$ matrix-matrix solve, enabling Tensor Core execution while preserving independent channel states inside the head.

---

**Algorithm 5** Block Two-Pass Algorithm (GPU Implementation)

---

**Require:** Block size $\ell$, inputs $\boldsymbol{u} \in \mathbb{R}^{b\ell \times d}$, coefficients $\boldsymbol{a} \in \mathbb{R}^{b\ell}$
**Ensure:** Approximate states $\tilde{\boldsymbol{x}} \in \mathbb{R}^{b\ell \times d}$
 1: **for all** blocks $t \in [b]$ **in parallel** assigned to warp $t$ **do**
 2:     **On chip do:**
 3:     Materialize $\boldsymbol{L}_t, \boldsymbol{g}_t \in \mathbb{R}^{\ell \times \ell}$ from coefficients $\boldsymbol{a}_t$                      ▷ In registers/SMEM
 4:     Load input tile $\boldsymbol{u}_t \in \mathbb{R}^{\ell \times d}$ from global memory to SMEM
 5:     $\boldsymbol{w}_t \leftarrow \boldsymbol{L}_t \boldsymbol{u}_t$                             ▷ Matrix multiplication via `wmma`
 6:     $v_t \leftarrow \boldsymbol{w}_{t,\ell} \in \mathbb{R}^{1 \times d}$              ▷ Extract carrier from last row of $\boldsymbol{w}_t$
 7:     Write $v_t$ to SMEM/DSMEM
 8:     Synchronize warps within CTA/cluster
 9:     **if** $t > 1$ **then**
10:         Read $v_{t-1}$ from SMEM/DSMEM
11:         $\tilde{\boldsymbol{x}}_t \leftarrow \boldsymbol{w}_t + \boldsymbol{g}_t v_{t-1}$                    ▷ Apply rank-1 update
12:     **else**
13:         $\tilde{\boldsymbol{x}}_t \leftarrow \boldsymbol{w}_t$
14:     **end if**
15:     Write block output $\tilde{\boldsymbol{x}}_t$ to global memory
16: **end for**

---

The resulting implementation, detailed in Algorithm 5, is actually a single-pass kernel from the perspective of global memory. Each warp materializes its local operators $\boldsymbol{L}_t$ and $\boldsymbol{g}_t$ on-the-fly from the input coefficients, computes the local solve $\boldsymbol{w}_t$, and extracts the carrier $v_t$. The carrier is passed to the next warp via SMEM or DSMEM, which then computes its final state. This pipelined dataflow ensures that inputs are read from global memory only once and outputs are written only once, while all intermediate carrier traffic remains on-chip, minimizing expensive off-chip memory access.

**Implementation details.** We implement the warp-level operations using CUTLASS, configuring a 32-thread `MmaTensorOp` for a $16 \times 16 \times 16$ tile shape. The operator acts on $\boldsymbol{L}_t$ (fp16/bf16), $\boldsymbol{u}_t$ (fp16/bf16), and accumulates into $\boldsymbol{w}_t$ using fp32 for precision. The on-chip extent of this approach is significant: a CTA with 32 warps can process $32 \times 16 = 512$ time steps, and a cluster of 16 CTAs on an H100 can process up to $16 \times 512 = 8192$ time steps without any global memory synchronization. For sequences exceeding this length, we segment the computation, checkpointing only the carrier vector between segments.

## 5 Layer Design

We introduce **Phalanx** layers for language modeling built using the Sliding Window Recurrence (SWR) sequence mixing primitive from Section 4. These layers exemplify how the Block Two-Pass algorithm can be instantiated as efficient local sequence mixers that complement global attention layers in hybrid architectures.

### 5.1 Parametrization Space and Design Philosophy

A parametrization of linear recurrences into a neural network layer requires several design choices: how to generate the recurrence coefficients $a_i$, whether to include gating mechanisms, how to share parameters across feature dimensions, and what activation functions to employ to bound the recurrence coefficients. The space of possible parametrizations is vast, Yang et al. (2024) provides a comprehensive taxonomy of linear recurrence variants, ranging from complex featurizers with polynomial expansions to sophisticated gating schemes.

We deliberately adopt a minimalist approach. Rather than exploring the full parametrization space, we focus on demonstrating that the SWR mixer itself, with its hardware-aligned block structure and local computation, provides a good efficiency-quality trade-off. Our design choices prioritize simplicity and hardware efficiency:

    *i.* Linear projections for all feature groups.

    *ii.* Sigmoid activation for projecting the recurrence coefficients $a_i$ and key-like gate $k_i$ to the stable range $[0, 1]$.

    *iii.* Head-based parameter sharing aligned with the $16 \times 16$ tile structure discussed in Section 4.2.2.

We maintain independent heads where each head feature evolves its own state with shared coefficients and no head feature mixing occurs during the recurrence computation. This structure (also used in recent works (Dao, 2024; Ku et al., 2025)) maps directly onto our tensor core model for the SWR mixer, in contrast to architectures like GLA and MultiHyena (Massaroli et al., 2023; Yang et al., 2023) that employ more complex head interactions leading to a matrix-valued state that requires a different adaptation of our kernel.

## 5.2 The Phalanx Layer

**Notation.** We adopt Einstein summation convention where repeated indices imply summation. Given an input sequence $u \in \mathbb{R}^{n \times D}$ where $D$ is the model dimension, we decompose computations across $h$ heads, each managing $d = D/h$ channels. Throughout, we use roman subscripts $(i, j, t)$ for position/time indices, Greek superscripts $(\alpha, \beta, \mu, \nu)$ for channel indices, and $\eta$ for head indices. Thus $u_i^{\eta\mu}$ denotes channel $\mu$ in head $\eta$ at position $i$, and expressions like $q_i^{\eta\nu} k_j^{\eta\nu}$ indicate summation over the repeated index $\nu$.

**Featurization.** The input is first projected to create the recurrence coefficients and gating features. Following the hardware considerations from Section 4.2.2, we organize computation into $h$ heads, where each head shares the same recurrence coefficients $a_i^{\eta}$ across its $d$ channels. Using Einstein notation (where repeated indices imply summation):

$$
\begin{aligned}
a_i^{\eta} &= \sigma(\boldsymbol{W}^{\eta\beta} u_i^{\beta}) && \text{(recurrence coefficient, sigmoid-bounded)} \\
q_i^{\eta\mu} &= \boldsymbol{Q}^{\eta\mu\beta} u_i^{\beta} && \text{(query-like gate)} \\
k_i^{\eta\mu} &= \sigma(\boldsymbol{K}^{\eta\mu\beta} u_i^{\beta}) && \text{(key-like gate)} \\
v_i^{\eta\mu} &= \boldsymbol{V}^{\eta\mu\beta} u_i^{\beta} && \text{(value)}
\end{aligned}
\tag{27}
$$

Here $\boldsymbol{W} \in \mathbb{R}^{h \times D}$ projects to head-wise recurrence coefficients, while $\boldsymbol{Q}, \boldsymbol{K}, \boldsymbol{V} \in \mathbb{R}^{h \times d \times D}$ are the standard attention-like projections[1]. The sigmoid activation ensures $a_i^{\eta} \in (0, 1)$, guaranteeing stability without additional regularization.

**Token and channel mixing.** We implement a simple SWR mixer with double gating:

$$
\begin{aligned}
\hat{u}_i^{\eta\mu} &= k_i^{\eta\mu} \cdot v_i^{\eta\mu} && \text{(pre-gate)} \\
x_i^{\eta\mu} &= \tilde{\boldsymbol{L}}_{ij}^{\eta} \hat{u}_j^{\eta\mu} && \text{(SWR (24) via B2P)} \\
y_i^{\eta\mu} &= q_i^{\eta\mu} \cdot x_i^{\eta\mu} + v_i^{\eta\mu} && \text{(post-gate with residual)}
\end{aligned}
\tag{28}
$$

where $\tilde{\boldsymbol{L}}^{\eta}$ is computed using the coefficients $\{a_i^{\eta}\}$. Finally, we apply an output projection to mix information across heads and produce the layer output:

$$
y_i^{\alpha} = \boldsymbol{O}^{\alpha\eta\mu} y_i^{\eta\mu} \quad \text{(output projection)}
\tag{29}
$$

with $\boldsymbol{O} \in \mathbb{R}^{D \times h \times d}$. Overall, the input-output mapping of the Phalanx layer can be compactly written as:

$$
y_i^{\alpha} = \boldsymbol{O}^{\alpha\eta\mu} \left[ (\boldsymbol{Q}^{\eta\mu\beta} u_i^{\beta}) \tilde{\boldsymbol{L}}_{ij}^{\eta} (\boldsymbol{K}^{\eta\nu\gamma} u_j^{\gamma})(\boldsymbol{V}^{\eta\nu\delta} u_j^{\delta}) + \boldsymbol{V}^{\eta\mu\beta} u_i^{\beta} \right]
\tag{30}
$$

---

[1]The three-dimensional structure of $\boldsymbol{Q}, \boldsymbol{K}, \boldsymbol{V}$ arises from absorbing the reshape operation into the linear projection: rather than first projecting $u_i^{\beta}$ to a flat $D$-dimensional vector and then reshaping to $(h, d)$, we directly parameterize the composed transformation. Since reshaping is itself a linear operation (a permutation of elements), this composition remains linear and can be represented as a single tensor contraction

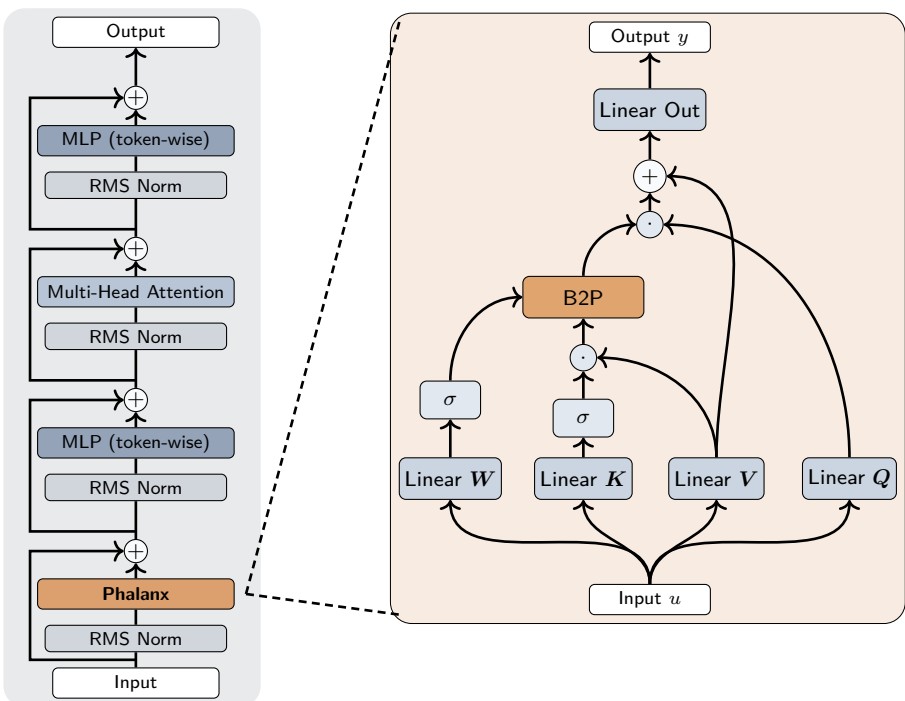

Figure 5.1: Hybrid architecture with Phalanx–MHA blocks (left) and Phalanx micro-architecture (right). The B2P mixer implements the SWR with pre-gate $k, v$ and post-gate $q$ projections.

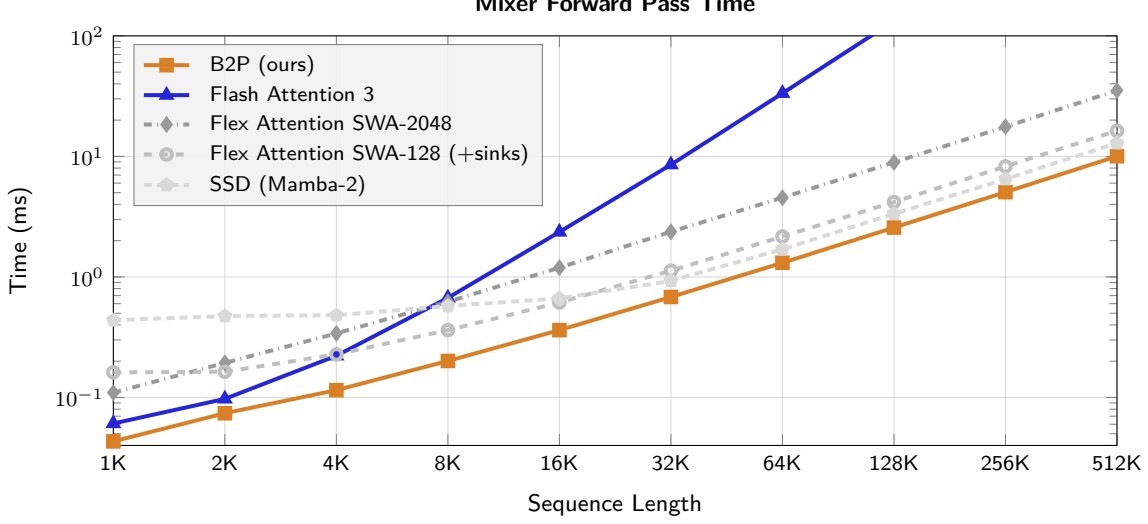

Figure 5.2: Phalanx has faster forward pass speed across sequence lengths ($D = 2048$, $h = 128$, $d = 16$, $\ell = 16$).

**Gate sharing across heads**   Modern transformer variants use strategies to share parameters and increase efficiency. One common strategy is GQA (Ainslie et al., 2023) to share key and values across groups of heads. Similar head sharing methods have been explored for recurrence layers (Dao, 2024). We apply group sharing separately to K and Q projections, such that within each group, multiple heads share the same gate parameters.

### 5.3 Integration in Hybrid Architectures

The Phalanx layer is designed as a building block for hybrid architectures, where it handles local sequence mixing while global attention layers capture long-range dependencies. This specialization of local recurrence for efficient local operations and global attention for longer range modeling enables scaling to long sequences without sacrificing quality.

### 5.4 Inference-time practicality

Phalanx layers retain the inference-time advantages of recurrent sequence mixers. During autoregressive decoding, the recurrent part of a Phalanx layer can be advanced from a compact per-layer state rather than an attention-style key-value cache that grows with context length. In a hybrid stack, the remaining full-attention layers still provide global retrieval and maintain their usual cache, while Phalanx replaces the local-mixer slots with constant-state streaming updates.

The main practical advantage appears during prefill. Full attention and global recurrent scans both introduce long-context data movement: attention materializes interactions against the prompt, while an exact hierarchical recurrence must propagate a carrier through all blocks. B2P avoids this global carrier solve. It computes dense local tiles and the nearest-neighbor correction in parallel, so the Phalanx prefill path stays bounded to the jagged local basis used during training.

The next section examines how these architectural choices translate to end-to-end performance in language modeling tasks.

## 6 Experiments

**Experimental Setup.** We compare Phalanx + Attention hybrids against strong Transformer and Sliding Window Attention (SWA) + Transformer hybrid baselines, focusing on perplexity and end-to-end training speed in the settings below.

At the largest scale, we train 1.3B parameter models for 100B tokens on FineWeb-Edu with identical settings. We use the Qwen3 tokenizer (vocab size 151,669), with 16 layers total and tied embeddings/output projections similar to Llama 3 (Dubey et al., 2024; Yang et al., 2025). Transformer layers, sliding window attention, use 16 heads and 8 heads for k and v, while Phalanx layers have a head size of 16 and 8 heads for k and v. Training uses a sequence length of 8192, batch size of 1M tokens, a peak learning rate of $3 \times 10^{-4}$ with 2B-token warmup, gradient clipping at 1.0, and cosine decay of the learning rate to 10% of the peak. We implement training in TorchTitan, extending it with features required for our setup. We use the `FlexAttention` (PyTorch Team, 2024) backend for SWA layers, while full-attention layers use the Flash Attention (Dao, 2023) backend for SDP. We further introduce a Phalanx-multihybrid model, which has alternating blocks of [Phalanx, SWA-128 with sinks, Phalanx, Attention].

To compare throughput, we evaluate 1.3B models on H100 80 GB GPUs at sequence lengths 4096, 8192, and 16384, using micro batch sizes 4, 2, and 1 respectively. Figure 6.3 compares Transformer++, SWA-128 hybrids with attention sinks, Mamba-2 hybrids implemented with Flash Linear Attention, and the Phalanx 1:1, 3:1, and multihybrid variants. At 16K context, the Phalanx models are the only variants in this comparison that remain within the H100 memory budget.

**Comparing Phalanx variants.** Based on comparing Phalanx variants on 40B tokens, we progress with training our design for Phalanx. To enable Phalanx to be decoded in recurrence mode, we can enforce decay to be large by bounding the maximum value of the transfer matrix. To test this we run an ablation enforcing transfer matrix values of 0.8 or less by scaling the sigmoid parametrization. Based on the better performance of Phalanx over these alternatives (Appendix C), we further perform two ablations at 10B tokens to test a version without input varying learned decay per head and a version without any input or output gating. We find these have higher loss compared to our layer (Phalanx-fixed-decay: 2.764, Phalanx-no-gates 2.713, Phalanx 2.696 at 10B tokens). Based on these results, we proceed with Phalanx. We also explore grouping

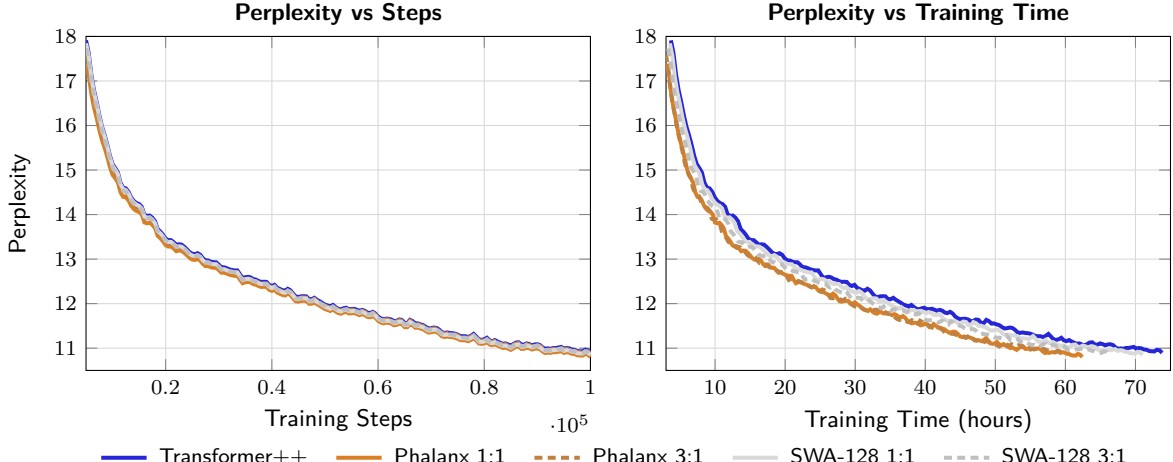

Figure 6.1: FineWeb-Edu Loss for Transformer++ and windowed hybrids

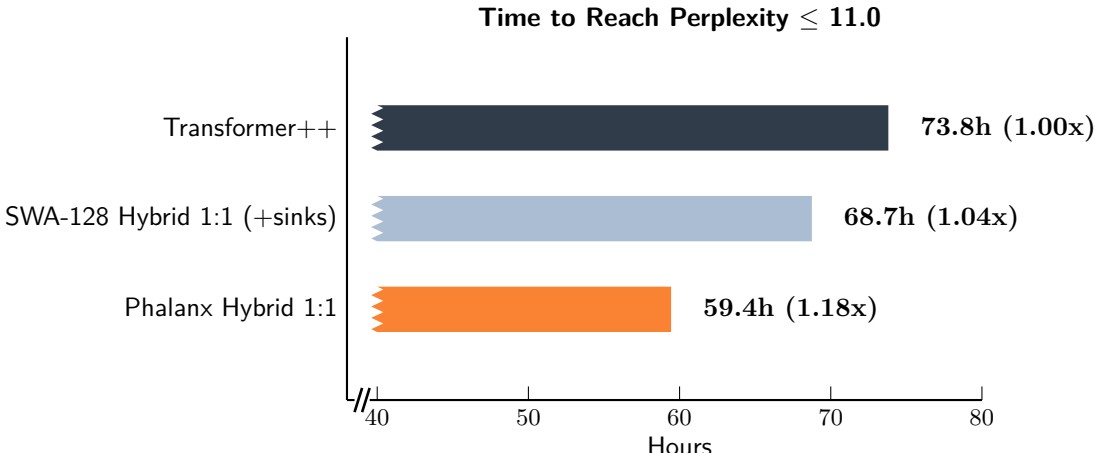

Figure 6.2: Wall-clock time to reach target perplexity on FineWeb-Edu.

heads for K and Q and found no deterioration when using 32, 16, or 8 groups for K and Q. We proceed with 8 groups for fair comparisons with other baselines.

**Training results.** As shown in Figure 6.1 and Figure 6.2, Phalanx-SWA-multihybrid achieves the highest overall efficiency in our comparison, training 24% faster than Transformer++ and 10% faster than sliding-window attention with comparable loss. At both 1:1 and 3:1 hybrid ratios, Phalanx achieves similar

| Model | Hybrid Ratio | Perplexity | Δ Perplexity | Train. k tok/s/GPU |
|---|---|---|---|---|
| Transformer++ | – | 10.95 | 0.00 | 49.4k |
| SWA-128 + sinks | 1:1 | 10.90 | −0.05 | 51.2k |
| SWA-2048 | 1:1 | 10.94 | −0.01 | 55.9k |
| SWA-128 | 1:1 | 16.07 | +5.12 | 60.4k |
| SWA-128 + sinks | 3:1 | 10.91 | −0.04 | 55.2k |
| Phalanx | 1:1 | 10.85 | −0.10 | 58.5k |
| Phalanx | 3:1 | 11.01 | +0.06 | 64.4k |
| Phalanx-SWA-multihybrid | 3:1 | 10.89 | −0.06 | 61.4k |

Table 6.1: Perplexity and training throughput comparison on B200 GPUs (8192 context length).

perplexity with 18% and 60% speedup against Transformers. Overall, the proposed Phalanx hybrids improve the quality–efficiency trade-off in these settings, delivering significantly improved training wall-clock time compared to the evaluated baselines while matching per-step loss. Downstream evaluation at this scale is reported in Appendices E and F, covering zero-shot benchmarks following Yang et al. (2024) and per-token-position perplexity up to 4× the training context. Phalanx+Attn (1:1) matches Transformer++ on standard benchmarks, while Phalanx-multihybrid reaches the Mamba-2 / Gated DeltaNet hybrid range.

**Sliding Window Attention.** The specific backend of sliding window attention (SWA) has a significant impact on both throughput and loss performance. Consistent with prior work (Agarwal et al., 2025; Wang et al., 2025a), we observe that attention sinks and scaling are critical for training short-context SWA layers. Without scaling, a window length of 128 leads to substantially degraded performance, with a +5.12 increase in perplexity compared to full attention. Incorporating attention sinks and scaling within `FlexAttention` improves the perplexity to -0.05 relative to the Transformer baseline. At larger window sizes, such as SWA-2048, model quality is comparable to full attention while being 13% faster.

### 6.1 Additional ablations

**Small-scale architecture sweep.** As a further sanity check we compare Phalanx hybrids against a broader set of recent efficient sequence models (Mamba-2 (Dao, 2024), Gated DeltaNet (Yang et al., 2024)) at 150M parameters on 5B tokens of Dolma-3, using a peak learning rate of $6 \times 10^{-4}$ and otherwise identical training recipes across architectures. Figure 6.4 places each model in the throughput/test-loss plane; Phalanx hybrids sit close to the Mamba-2 / GDN cluster on quality while training 10 to 30% faster on H100. Full downstream-evaluation results are reported in Appendix D.

**Learning-rate sweep.** To verify that our comparison does not penalise either architecture with a suboptimal learning rate, we sweep the peak LR at 340M parameters on 15B tokens of FineWeb-Edu. Schedule and other hyperparameters match the headline setup. Phalanx+Attention (1:1) reaches lower train and validation loss than Transformer++ at every learning rate in the sweep (Figure 6.5). Loss continues to improve up to $1.6 \times 10^{-3}$, the highest stable LR for both models; pushing to $3.2 \times 10^{-3}$ diverges training for Phalanx and Transformer++ alike. Since $8 \times 10^{-4}$ already sits near the plateau and leaves a stability margin under the LR shift induced by scaling up, we adopt it as the 340M operating point and $\mu$P-scale it to $4 \times 10^{-4}$ for the 1.3B runs, matching the GDN recipe (Yang et al., 2024).

## 7 Conclusion

We propose a framework for studying linear recurrences as matrices enabling us to connect them to modern hardware level. Based on these insights we introduce Sliding Window Recurrences to provide local, efficient sequence mixer operators. Bounded, block-structured recurrences are not intended to recover arbitrarily long-range dependencies in isolation; instead, they are designed to minimize long-range data movement across the GPU memory hierarchy while providing high-throughput token mixing. This enables hybrid models where attention handles global context and local context is handled by efficient mixers. Consequently, the key evaluation is how a layer contributes to end-to-end throughput, accuracy, and scaling when composed with complementary global modules like attention. We apply this paradigm to develop Phalanx hybrid models and demonstrate their efficiency and performance.

## Broader Impact Statement

Phalanx reduces the wall-clock cost of pretraining hybrid language models at a given quality target. This makes large-scale training more compute-efficient by reducing energy use and hardware time for a fixed experimental objective. More broadly, training-efficiency improvements can shift the cost frontier for language-model development, so their impact depends on the downstream systems in which they are incorporated.

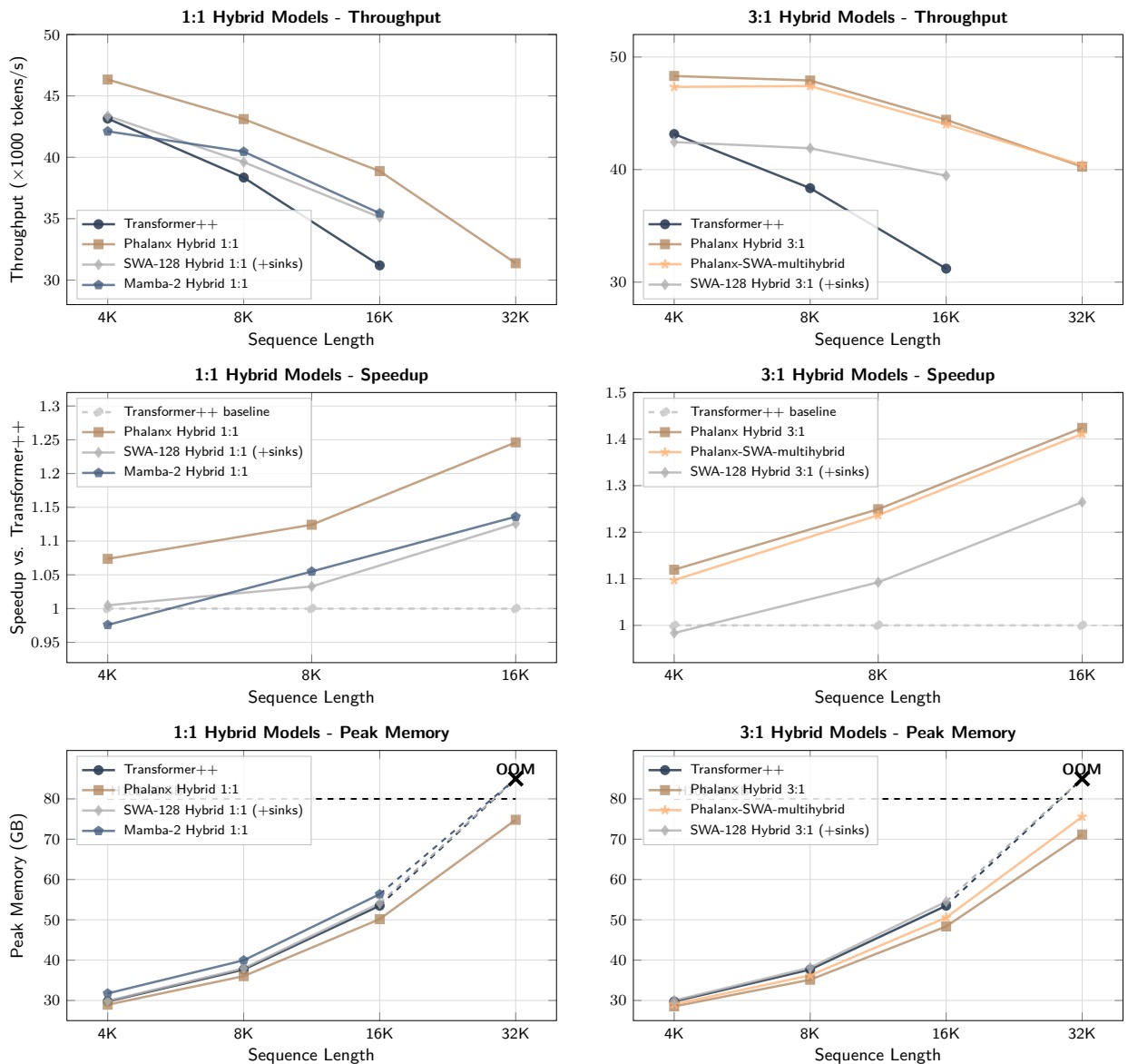

Figure 6.3: Training throughput, speedups, and peak memory across sequence lengths for hybrid models (1:1, 3:1) on H100 80 GB.

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

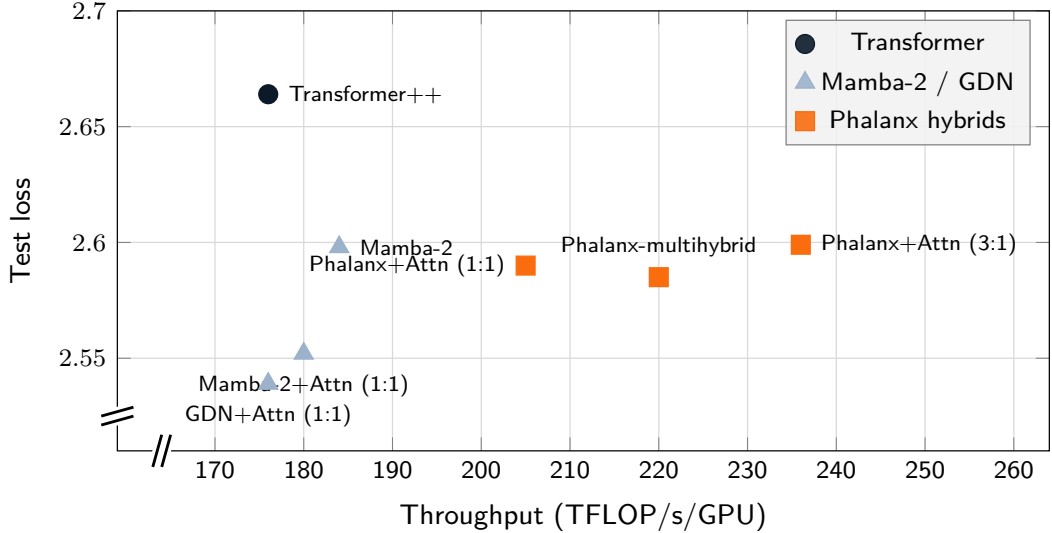

Figure 6.4: Training throughput vs. test loss at 150M / 5B tokens on Dolma-3.

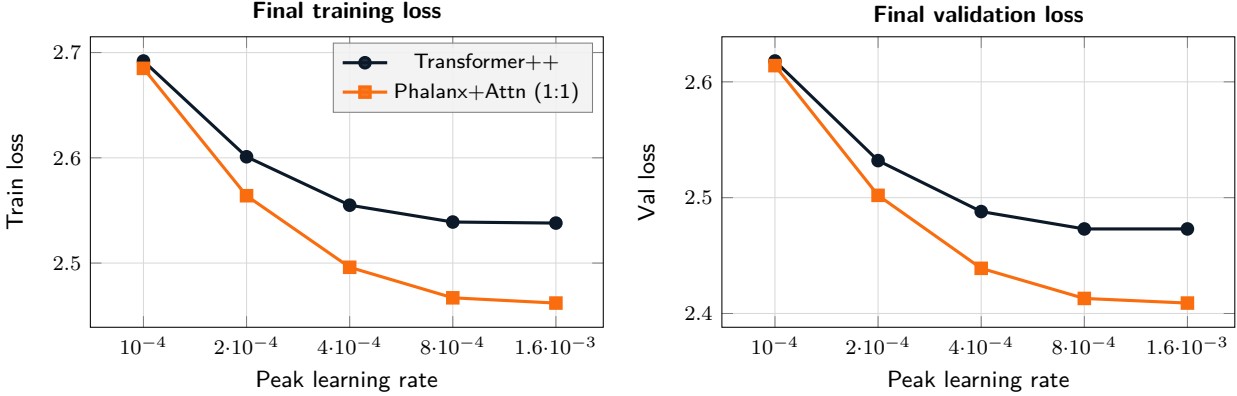

Figure 6.5: Phase-2 LR sweep at 340M / 15B FineWeb-Edu.

Iz Beltagy, Matthew E Peters, and Arman Cohan. Longformer: The long-document transformer. *arXiv preprint arXiv:2004.05150*, 2020.

Guy E Blelloch. Prefix sums and their applications. 1990.

Richard P. Brent and H. T. Kung. A regular layout for parallel adders. *IEEE Transactions on Computers*, 100(3):260–264, 1982.

Tom Brown, Benjamin Mann, Nick Ryder, Melanie Subbiah, Jared D Kaplan, Prafulla Dhariwal, Arvind Neelakantan, Pranav Shyam, Girish Sastry, Amanda Askell, et al. Language models are few-shot learners. *Advances in neural information processing systems*, 33:1877–1901, 2020.

Keshigeyan Chandrasegaran, Michael Poli, Daniel Y Fu, Dongjun Kim, Lea M Hadzic, Manling Li, Agrim Gupta, Stefano Massaroli, Azalia Mirhoseini, Juan Carlos Niebles, et al. Exploring diffusion transformer designs via grafting. *arXiv preprint arXiv:2506.05340*, 2025.

Shiv Chandrasekaran, Patrick Dewilde, Ming Gu, T Pals, Xiaorui Sun, Alle-Jan van der Veen, and Daniel White. Some fast algorithms for sequentially semiseparable representations. *SIAM Journal on Matrix Analysis and Applications*, 27(2):341–364, 2005.

Tri Dao. Flashattention-2: Faster attention with better parallelism and work partitioning. *arXiv preprint arXiv:2307.08691*, 2023.

Tri Dao. State space duality (mamba-2) part iii – the algorithm, 2024. URL https://tridao.me/blog/2024/mamba2-part3-algorithm/.

Tri Dao and Albert Gu. Transformers are ssms: Generalized models and efficient algorithms through structured state space duality. *arXiv preprint arXiv:2405.21060*, 2024.

P Dewilde and AJ van der Veen. Semi-and quasi-separable systems. In *Operator Theory*, pp. 901–930. Springer, 2014.

Patrick Dewilde and Alle-Jan Van der Veen. *Time-varying systems and computations.* Springer Science & Business Media, 1998.

Patrick Dewilde, Klaus Diepold, and Alle-Jan van der Veen. Time-variant and quasi-separable systems: matrix theory, recursions and computations. 2025.

Abhimanyu Dubey, Abhinav Jauhri, Abhinav Pandey, Abhishek Kadian, Ahmad Al-Dahle, Aiesha Letman, Akhil Mathur, Alan Schelten, Amy Yang, Angela Fan, et al. The llama 3 herd of models. *arXiv e-prints*, pp. arXiv–2407, 2024.

Leo Gao, Jonathan Tow, Baber Abbasi, Stella Biderman, Sid Black, Anthony DiPofi, Charles Foster, Laurence Golding, Jeffrey Hsu, Alain Le Noac'h, Haonan Li, Kyle McDonell, Niklas Muennighoff, Chris Ociepa, Jason Phang, Laria Reynolds, Hailey Schoelkopf, Aviya Skowron, Lintang Sutawika, Eric Tang, Anish Thite, Ben Wang, Kevin Wang, and Andy Zou. The language model evaluation harness, 07 2024. URL https://zenodo.org/records/12608602.

I Gohberg, MA Kaashoek, and L Lerer. Minimality and realization of discrete time-varying systems. In *Time-variant systems and interpolation*, pp. 261–296. Springer, 1992.

Albert Gu and Tri Dao. Mamba: Linear-time sequence modeling with selective state spaces. *arXiv preprint arXiv:2312.00752*, 2023.

Mark Harris, Shubhabrata Sengupta, and John D. Owens. Parallel prefix sum (scan) with CUDA. In Hubert Nguyen (ed.), *GPU Gems 3*, chapter 39, pp. 851–876. Addison-Wesley Professional, 2007.

David E Keyes, Hatem Ltaief, and George Turkiyyah. Hierarchical algorithms on hierarchical architectures. *Philosophical Transactions of the Royal Society A*, 378(2166):20190055, 2020.

Peter M. Kogge and Harold S. Stone. A parallel algorithm for the efficient solution of a general class of recurrence equations. *IEEE Transactions on Computers*, 100(8):786–793, 2009.

Jerome Ku, Eric Nguyen, David W Romero, Garyk Brixi, Brandon Yang, Anton Vorontsov, Ali Taghibakhshi, Amy X Lu, Dave P Burke, Greg Brockman, et al. Systems and algorithms for convolutional multi-hybrid language models at scale. *arXiv preprint arXiv:2503.01868*, 2025.

Eric Martin and Chris Cundy. Parallelizing linear recurrent neural nets over sequence length. *arXiv preprint arXiv:1709.04057*, 2017.

Stefano Massaroli, Michael Poli, Dan Fu, Hermann Kumbong, Rom Parnichkun, David Romero, Aman Timalsina, Quinn McIntyre, Beidi Chen, Atri Rudra, et al. Laughing hyena distillery: Extracting compact recurrences from convolutions. *Advances in Neural Information Processing Systems*, 36:17072–17116, 2023.

Duane Merrill and Michael Garland. Single-pass parallel prefix scan with decoupled look-back. *NVIDIA, Tech. Rep. NVR-2016-002*, 2016.

Michael Poli, Stefano Massaroli, Eric Nguyen, Daniel Y Fu, Tri Dao, Stephen Baccus, Yoshua Bengio, Stefano Ermon, and Christopher Ré. Hyena hierarchy: Towards larger convolutional language models. In *International Conference on Machine Learning*, pp. 28043–28078. PMLR, 2023.

Michael Poli, Armin W Thomas, Eric Nguyen, Pragaash Ponnusamy, Björn Deiseroth, Kristian Kersting, Taiji Suzuki, Brian Hie, Stefano Ermon, Christopher Ré, et al. Mechanistic design and scaling of hybrid architectures. *arXiv preprint arXiv:2403.17844*, 2024.

PyTorch Team. Flexattention: The flexibility of pytorch with the performance of flashattention. https://pytorch.org/blog/flexattention/, November 2024. PyTorch Blog.

David W Romero, Anna Kuzina, Erik J Bekkers, Jakub M Tomczak, and Mark Hoogendoorn. Ckconv: Continuous kernel convolution for sequential data. *arXiv preprint arXiv:2102.02611*, 2021.

Jimmy TH Smith, Andrew Warrington, and Scott W Linderman. Simplified state space layers for sequence modeling. *arXiv preprint arXiv:2208.04933*, 2022.

Armin W Thomas, Rom Parnichkun, Alexander Amini, Stefano Massaroli, and Michael Poli. Star: Synthesis of tailored architectures. *arXiv preprint arXiv:2411.17800*, 2024.

Raf Vandebril, Marc Van Barel, and Nicola Mastronardi. *Matrix computations and semiseparable matrices: linear systems*, volume 1. JHU Press, 2008.

Johannes von Oswald, Nino Scherrer, Seijin Kobayashi, Luca Versari, Songlin Yang, Maximilian Schlegel, Kaitlin Maile, Yanick Schimpf, Oliver Sieberling, Alexander Meulemans, Rif A. Saurous, Razvan Pascanu, and João Sacramento. MesaNet: Sequence modeling by locally optimal test-time training. In *International Conference on Learning Representations (ICLR)*, 2026. arXiv:2506.05233.

Bailin Wang, Chang Lan, Chong Wang, and Ruoming Pang. Rattention: Towards the minimal sliding window size in local-global attention models. *arXiv preprint arXiv:2506.15545*, 2025a.

Dustin Wang, Rui-Jie Zhu, Steven Abreu, Yong Shan, Taylor Kergan, Yuqi Pan, Yuhong Chou, Zheng Li, Ge Zhang, Wenhao Huang, et al. A systematic analysis of hybrid linear attention. *arXiv preprint arXiv:2507.06457*, 2025b.

An Yang, Anfeng Li, Baosong Yang, Beichen Zhang, Binyuan Hui, Bo Zheng, Bowen Yu, Chang Gao, Chengen Huang, Chenxu Lv, et al. Qwen3 technical report. *arXiv preprint arXiv:2505.09388*, 2025.

Songlin Yang, Bailin Wang, Yikang Shen, Rameswar Panda, and Yoon Kim. Gated linear attention transformers with hardware-efficient training. *arXiv preprint arXiv:2312.06635*, 2023.

Songlin Yang, Jan Kautz, and Ali Hatamizadeh. Gated delta networks: Improving mamba2 with delta rule. *arXiv preprint arXiv:2412.06464*, 2024.

Manzil Zaheer, Guru Guruganesh, Kumar Avinava Dubey, Joshua Ainslie, Chris Alberti, Santiago Ontanon, Philip Pham, Anirudh Ravula, Qifan Wang, Li Yang, et al. Big bird: Transformers for longer sequences. *Advances in neural information processing systems*, 33:17283–17297, 2020.

Michael Zhang, Simran Arora, Rahul Chalamala, Alan Wu, Benjamin Spector, Aaryan Singhal, Krithik Ramesh, and Christopher Ré. Lolcats: On low-rank linearizing of large language models. *arXiv preprint arXiv:2410.10254*, 2024.

# A  Algebraic Properties of Weighted Shifts

Let $\mathbb{F}$ be a field, and let $\boldsymbol{M} \in \mathbb{F}^{n \times n}$ be a strictly lower triangular matrix, i.e., $\boldsymbol{M}_{ij} = 0$ whenever $i \leq j$. Let $\{e_i\}_{i=1}^n$ denote the standard basis of $\mathbb{F}^n$. We define the standard descending flag of subspaces:

$$\mathbb{F}^n = U_1 \supset U_2 \supset \cdots \supset U_n \supset U_{n+1} = \{0\},$$

where $U_k := \text{span}\{e_i\}_{i=k}^n$. Note that $\dim(U_k) = n - k + 1$.

**Lemma 1** (Nilpotency of strictly lower triangular matrices). *If $\boldsymbol{M} \in \mathbb{F}^{n \times n}$ is strictly lower triangular, then $\boldsymbol{M}^n = \boldsymbol{0}$.*

*Proof.* We prove that $M(U_k) \subseteq U_{k+1}$ for all $k = 1, \ldots, n$ by examining the action of $M$ on the basis vectors of $U_k$. If $e_j \in U_k$, then $j \geq k$. The vector $Me_j$ corresponds to the $j$-th column of $M$:

$$Me_j = \sum_{i=1}^{n} M_{ij} e_i.$$

Since $M$ is strictly lower triangular, $M_{ij} = 0$ if $i \leq j$. Thus, the summation simplifies to:

$$Me_j = \sum_{i=j+1}^{n} M_{ij} e_i.$$

This resulting vector is in the span of $\{e_{j+1}, \ldots, e_n\}$, so $Me_j \in U_{j+1}$. Since $j \geq k$, we have $j + 1 \geq k + 1$, which implies $U_{j+1} \subseteq U_{k+1}$. By linearity, $M(U_k) \subseteq U_{k+1}$. Applying this iteratively to the entire space $\mathbb{F}^n = U_1$ gives:

$$M^n(U_1) \subseteq M^{n-1}(U_2) \subseteq M^{n-2}(U_3) \subseteq \cdots \subseteq M(U_n) \subseteq U_{n+1}.$$

Since $U_{n+1} = \{0\}$, we have $M^n(\mathbb{F}^n) = \{0\}$. Therefore, $M^n$ must be the zero matrix. $\square$

A crucial consequence of a matrix being nilpotent is that its subtraction from the identity matrix yields an invertible matrix.

**Corollary A.1** (Invertibility of $I - M$). *If $M \in \mathbb{F}^{n \times n}$ is strictly lower triangular, then the matrix $I - M$ is invertible.*

*Proof.* By Lemma 1, $M$ is nilpotent with $M^n = 0$. We can explicitly construct the inverse using a finite geometric series. Let:

$$S = I + M + M^2 + \cdots + M^{n-1}.$$

We verify that $S$ is the inverse of $I - M$:

$$\begin{aligned}
(I - M)S &= (I - M)(I + M + \cdots + M^{n-1}) \\
&= (I + M + \cdots + M^{n-1}) - (M + M^2 + \cdots + M^{n-1} + M^n) \\
&= I - M^n.
\end{aligned}$$

Since $M^n = 0$, we have $(I - M)S = I$. Similarly, $S(I - M) = I$. Thus, $I - M$ is invertible with $(I - M)^{-1} = S$. $\square$

**The case of weighted shift matrices $AZ$.** A concrete illustration of this theory is provided by the weighted shift matrices with which we construct the transfer operator of scalar linear recurrences. Let $Z$ be the down-shift matrix on $\mathbb{R}^n$ with entries $Z_{ij} = \delta_{i,j+1}$ (ones on the first subdiagonal), and let $A = \mathrm{diag}(a_1, \ldots, a_n)$ be the diagonal matrix of coefficients. The weighted shift $AZ$ has entries $(AZ)_{ij} = a_i Z_{ij}$, resulting in the following structure:

$$AZ = \begin{pmatrix}
0 & 0 & \cdots & 0 & 0 \\
a_2 & 0 & \cdots & 0 & 0 \\
0 & a_3 & \cdots & 0 & 0 \\
\vdots & \vdots & \ddots & \vdots & \vdots \\
0 & 0 & \cdots & a_n & 0
\end{pmatrix}.$$

The matrix $AZ$ is strictly lower triangular, and by Lemma 1, $(AZ)^n = 0$.

The action of $AZ$ on the standard basis visualizes the movement down the flag. For $j = 1, \ldots, n - 1$, $(AZ)e_j = a_{j+1} e_{j+1}$ and $(AZ)e_n = \mathbb{0}_n$.

$$e_1 \xrightarrow{a_2} e_2 \xrightarrow{a_3} e_3 \to \cdots \to e_{n-1} \xrightarrow{a_n} e_n \to \mathbb{0}_n.$$

For $k \geq 0$ and $j \in [n]$,

$$(\boldsymbol{AZ})^k e_j = a_{j+k:j+1} \cdot e_{j+k}, \qquad (\text{where } e_m := \mathbb{0}_n \text{ if } m > n),$$

using the product notation $a_{i:j} = a_i a_{i-1} \cdots a_j$ from Section 3. In particular, $(\boldsymbol{AZ})^{n-1} e_1 = a_{n:2} \cdot e_n$. The nilpotency index $m$ (the smallest integer with $(\boldsymbol{AZ})^m = \mathbf{0}$) satisfies $m \leq n$. Equality $m = n$ occurs if and only if $a_2, \ldots, a_n$ are all nonzero. If the weights allow for a shorter path to zero (i.e., some $a_{k+1} = 0$), the index is smaller. More generally, if the longest contiguous block of nonzero weights among $(a_2, \ldots, a_n)$ has length $L$, then $m = L + 1$.

**Connection to the transfer operator.** The results above provide the algebraic foundation for the transfer operator $\boldsymbol{L} = (\boldsymbol{I} - \boldsymbol{AZ})^{-1}$ in Section 3. By Corollary A.1 (with $\boldsymbol{M} = \boldsymbol{AZ}$), the matrix $\boldsymbol{I} - \boldsymbol{AZ}$ is always invertible, ensuring that the linear system (2) has a unique solution. Moreover, the nilpotency of $\boldsymbol{AZ}$ guarantees that the Neumann series expansion (3) terminates after finitely many terms, yielding an exact representation rather than an infinite series. This finite expansion is crucial for both the theoretical analysis and the practical computation of the transfer operator, enabling the explicit characterization of its entries as $\boldsymbol{L}_{ij} = a_{i:j+1}$ and facilitating the derivation of efficient parallel algorithms.

## B Three Proofs of the Binary Factorization of Geometric Series

The binary factorization of the geometric series is fundamental to parallel algorithms for linear recurrences, particularly the Kogge-Stone algorithm (Kogge & Stone, 2009) discussed in the main text. This identity shows how a sum of matrix powers can be factorized into a product of sparse matrices, enabling logarithmic-depth parallel computation. While the identity is algebraic and holds in any ring, its application to the transfer operator $(\boldsymbol{I} - \boldsymbol{AZ})^{-1}$ of linear recurrences reveals deep connections between parallel algorithms and matrix factorizations. We present three proofs that illuminate different aspects of this remarkable identity: the first uses telescoping cancellation, the second employs mathematical induction, and the third exploits the binary expansion of integers.

**Proposition B.1** (Binary Factorization of Geometric Series). *Let $\boldsymbol{M} \in \mathbb{R}^{n \times n}$ be any matrix and let $n = 2^m$ for some integer $m \geq 1$. The following algebraic identity holds:*

$$\sum_{k=0}^{n-1} \boldsymbol{M}^k = \prod_{j=0}^{m-1} (\boldsymbol{I} + \boldsymbol{M}^{2^j}). \tag{31}$$

*If $(\boldsymbol{I} - \boldsymbol{M})$ is invertible, the sum is the partial sum of the Neumann series, and both sides are equal to $(\boldsymbol{I} - \boldsymbol{M}^n)(\boldsymbol{I} - \boldsymbol{M})^{-1}$. In the special case where $\boldsymbol{M}$ is nilpotent with $\boldsymbol{M}^n = 0$, this simplifies to $(\boldsymbol{I} - \boldsymbol{M})^{-1}$.*

*Proof 1 (Difference of Squares).* This proof relies on expressing the term $\boldsymbol{I} - \boldsymbol{M}^n$ in two different ways. First, by repeatedly applying the difference of squares formula we can expand $\boldsymbol{I} - \boldsymbol{M}^n = \boldsymbol{I} - \boldsymbol{M}^{2^m}$ as a product:

$$\boldsymbol{I} - \boldsymbol{M}^{2^m} = (\boldsymbol{I} + \boldsymbol{M}^{2^{m-1}})(\boldsymbol{I} - \boldsymbol{M}^{2^{m-1}}) \tag{32}$$

$$= (\boldsymbol{I} + \boldsymbol{M}^{2^{m-2}})(\boldsymbol{I} + \boldsymbol{M}^{2^{m-2}})(\boldsymbol{I} - \boldsymbol{M}^{2^{m-1}}) \tag{33}$$

$$\vdots$$

$$= (\boldsymbol{I} + \boldsymbol{M}^{2^{m-1}}) \cdots (\boldsymbol{I} + \boldsymbol{M}^2)(\boldsymbol{I} + \boldsymbol{M})(\boldsymbol{I} - \boldsymbol{M}) \tag{34}$$

$$= \prod_{j=0}^{m-1} (\boldsymbol{I} + \boldsymbol{M}^{2^j})(\boldsymbol{I} - \boldsymbol{M}). \tag{35}$$

Second, the standard summation formula for a finite geometric series gives the same term:

$$\boldsymbol{I} - \boldsymbol{M}^n = \sum_{k=0}^{n-1} \boldsymbol{M}^k (\boldsymbol{I} - \boldsymbol{M}). \tag{36}$$

By equating the two expressions for $\boldsymbol{I} - \boldsymbol{M}^n$, we have

$$\prod_{j=0}^{m-1}(\boldsymbol{I} + \boldsymbol{M}^{2^j})(\boldsymbol{I} - \boldsymbol{M}) = \sum_{k=0}^{n-1}\boldsymbol{M}^k(\boldsymbol{I} - \boldsymbol{M}). \tag{37}$$

The identity (31) holds in the ring of polynomials in $\boldsymbol{M}$. If $(\boldsymbol{I} - \boldsymbol{M})$ is invertible, we can multiply both sides by $(\boldsymbol{I} - \boldsymbol{M})^{-1}$ to establish the equality, but the underlying identity is purely algebraic and holds regardless. $\square$

*Proof 2 (Induction).* We prove the algebraic identity (31) by induction. For any $t \geq 1$, let $S_t(\boldsymbol{M}) = \sum_{k=0}^{2^t-1}\boldsymbol{M}^k$ and $P_t(\boldsymbol{M}) = \prod_{j=0}^{t-1}(\boldsymbol{I} + \boldsymbol{M}^{2^j})$. We show that $S_t(\boldsymbol{M}) = P_t(\boldsymbol{M})$ for all $t \geq 1$. **Base case:** For $t = 1$, we have $2^t = 2$.

$$S_1(\boldsymbol{M}) = \boldsymbol{I} + \boldsymbol{M} \tag{38}$$

$$P_1(\boldsymbol{M}) = \boldsymbol{I} + \boldsymbol{M}^{2^0} = \boldsymbol{I} + \boldsymbol{M}. \tag{39}$$

The identity holds. **Inductive step:** Assume the identity $S_t(\boldsymbol{M}) = P_t(\boldsymbol{M})$ holds for some $t \geq 1$. We seek to prove it for $t + 1$.

$$S_{t+1}(\boldsymbol{M}) = \sum_{k=0}^{2^{t+1}-1}\boldsymbol{M}^k = \sum_{k=0}^{2^t-1}\boldsymbol{M}^k + \sum_{k=2^t}^{2^{t+1}-1}\boldsymbol{M}^k \tag{40}$$

$$= S_t(\boldsymbol{M}) + \boldsymbol{M}^{2^t}\sum_{l=0}^{2^t-1}\boldsymbol{M}^l \tag{41}$$

$$= S_t(\boldsymbol{M}) + \boldsymbol{M}^{2^t}S_t(\boldsymbol{M}) = (\boldsymbol{I} + \boldsymbol{M}^{2^t})S_t(\boldsymbol{M}). \tag{42}$$

Using the inductive hypothesis $S_t(\boldsymbol{M}) = P_t(\boldsymbol{M})$, we get

$$S_{t+1}(\boldsymbol{M}) = (\boldsymbol{I} + \boldsymbol{M}^{2^t})P_t(\boldsymbol{M}) = (\boldsymbol{I} + \boldsymbol{M}^{2^t})\prod_{j=0}^{t-1}(\boldsymbol{I} + \boldsymbol{M}^{2^j}) = \prod_{j=0}^{t}(\boldsymbol{I} + \boldsymbol{M}^{2^j}) = P_{t+1}(\boldsymbol{M}). \tag{43}$$

This completes the induction. Setting $t = m$ gives the desired result. $\square$

*Proof 3 (Binary Expansion).* We expand the product on the right-hand side of (31). Each term in the expansion corresponds to a unique choice of either $\boldsymbol{I}$ or $\boldsymbol{M}^{2^j}$ for each factor $j \in \{0, \ldots, m-1\}$. We can represent such a choice by a binary vector $\mathbf{b} = (b_0, b_1, \ldots, b_{m-1}) \in \{0, 1\}^m$. The expansion is a sum over all $2^m$ possible choices of $\mathbf{b}$:

$$\prod_{j=0}^{m-1}(\boldsymbol{I} + \boldsymbol{M}^{2^j}) = \sum_{\mathbf{b}\in\{0,1\}^m}\prod_{j=0}^{m-1}(\boldsymbol{M}^{2^j})^{b_j} = \sum_{\mathbf{b}\in\{0,1\}^m}\boldsymbol{M}^{\sum_{j=0}^{m-1}b_j2^j}. \tag{44}$$

The map from a binary vector $\mathbf{b}$ to the integer $k = \sum_{j=0}^{m-1}b_j2^j$ is a bijection from $\{0, 1\}^m$ to the set of integers $\{0, 1, \ldots, 2^m - 1\}$. Therefore, summing over all possible binary vectors $\mathbf{b}$ is equivalent to summing over all integer powers of $\boldsymbol{M}$ from $k = 0$ to $k = 2^m - 1$:

$$\prod_{j=0}^{m-1}(\boldsymbol{I} + \boldsymbol{M}^{2^j}) = \sum_{\mathbf{b}\in\{0,1\}^m}\prod_{j=0}^{m-1}(\boldsymbol{M}^{2^j})^{b_j} = \sum_{\mathbf{b}\in\{0,1\}^m}\boldsymbol{M}^{\sum_{j=0}^{m-1}b_j2^j}. \tag{45}$$

The map from a binary vector $\mathbf{b}$ to the integer $k = \sum_{j=0}^{m-1}b_j2^j$ is a bijection from $\{0, 1\}^m$ to the set of integers $\{0, 1, \ldots, 2^m - 1\}$. Therefore, summing over all possible binary vectors $\mathbf{b}$ is equivalent to summing over all integer powers of $\boldsymbol{M}$ from $k = 0$ to $k = 2^m - 1$:

$$\prod_{j=0}^{m-1}(\boldsymbol{I} + \boldsymbol{M}^{2^j}) = \sum_{k=0}^{2^m-1}\boldsymbol{M}^k. \tag{46}$$

$\square$

## C  Additional Phalanx Ablation

| Ablations | Loss at 40B tokens |
|---|---|
| Transformer++ | 2.52 |
| Phalanx | 2.51 |
| Phalanx-decay | 2.56 |

Table C.1: Comparing loss of Phalanx with set minimal decay (transfer matrix maximum of 0.8)

## D  Small-scale architecture sweep: downstream evaluation

Complement to the 150M scatter in Section 6.1. All models are trained for 5B tokens on Dolma-3 with sequence length 4096, global batch 128, peak learning rate $6 \times 10^{-4}$ with cosine decay to lr/10, identical across architectures. Attention slots in the `h1/h3` hybrids and in Mamba-2+Attn / GDN+Attn are plain full causal attention; only Phalanx-multihybrid applies SWA-128 with per-head attention sinks (via FlexAttention). Evaluation uses EleutherAI's `lm-evaluation-harness` (Gao et al., 2024); we report accuracy-normalised scores where available and perplexity otherwise.

| Mixer | Test loss↓ | Wiki PPL↓ | LMB PPL↓ | LMB acc↑ | PIQA↑ | HS†↑ | Wino†↑ | ARC-e↑ | ARC-c†↑ | BoolQ↑ |
|---|---|---|---|---|---|---|---|---|---|---|
| Transformer++ | 2.664 | 49.0 | 80.7 | 25.9 | 59.6 | 28.4 | 48.8 | 38.2 | 22.9 | 62.1 |
| Mamba-2 | 2.598 | 45.0 | 65.8 | 24.8 | 60.5 | 29.6 | 50.6 | 39.3 | 23.7 | 61.5 |
| Mamba-2+Attn (1:1) | 2.552 | 41.3 | 57.9 | 27.7 | 59.1 | 29.6 | 49.6 | 40.1 | 24.9 | 52.3 |
| GDN+Attn (1:1) | 2.539 | 40.3 | 58.9 | 27.1 | 59.7 | 29.8 | 51.2 | 40.1 | 22.3 | 60.0 |
| Phalanx+Attn (1:1) | 2.590 | 43.8 | 63.9 | 27.8 | 59.1 | 29.5 | 49.6 | 40.0 | 22.9 | 59.7 |
| Phalanx+Attn (3:1) | 2.599 | 44.6 | 68.1 | 25.6 | 59.8 | 29.1 | 48.3 | 38.7 | 24.0 | 60.2 |
| Phalanx-multihybrid | 2.585 | 43.8 | 66.7 | 26.4 | 59.6 | 29.5 | 51.9 | 39.7 | 22.6 | 60.9 |

Table D.1: Small-scale architecture sweep at 150M parameters / 5B tokens on Dolma-3.

## E  Zero-shot evaluation at 1.3B

Complement to Section 6. We evaluate two 1.3B Phalanx checkpoints, Phalanx+Attn (1:1) and Phalanx-multihybrid, against the recurrent and hybrid baselines reported in Table 3 of Yang et al. (2024). All baselines were trained for 100B tokens under that paper's recipe; we train our two Phalanx hybrids end-to-end with the identical recipe so the rows are directly comparable without re-baselining. Evaluation uses EleutherAI's `lm-evaluation-harness` (Gao et al., 2024); PIQA, WinoGrande, ARC-easy, and BoolQ report accuracy, while HellaSwag and ARC-challenge report length-normalised accuracy, matching the metric conventions of Yang et al. (2024).

| Mixer | Wiki PPL↓ | LMB PPL↓ | LMB acc↑ | PIQA↑ | HS↑ | Wino↑ | ARC-e↑ | ARC-c↑ | BoolQ↑ |
|---|---|---|---|---|---|---|---|---|---|
| RetNet | 19.08 | 17.27 | 40.52 | 70.07 | 49.16 | 54.14 | 67.34 | 33.78 | 60.39 |
| HGRN2 | 19.10 | 17.69 | 39.54 | 70.45 | 49.53 | 52.80 | 69.40 | 35.32 | 56.66 |
| Mamba | 17.92 | 15.06 | 43.98 | 71.32 | 52.91 | 52.95 | 69.52 | 35.40 | 61.13 |
| Mamba-2 | 16.56 | 12.56 | 45.66 | 71.87 | 55.67 | 55.24 | 72.47 | 37.88 | 60.13 |
| DeltaNet | 17.71 | 16.88 | 42.46 | 70.72 | 50.93 | 53.35 | 68.47 | 35.66 | 55.29 |
| Gated DeltaNet | 16.42 | 12.17 | 46.65 | 72.25 | 55.76 | 57.45 | 71.21 | 38.39 | 60.24 |
| Transformer++ | 18.53 | 18.32 | 42.60 | 70.02 | 50.23 | 53.51 | 68.83 | 35.10 | 57.09 |
| Samba | 16.13 | 13.29 | 44.94 | 70.94 | 53.42 | 55.56 | 68.81 | 36.17 | 62.11 |
| GDN-H1 | 16.07 | 12.12 | 47.73 | 72.57 | 56.53 | 58.40 | 71.75 | 40.10 | 63.21 |
| GDN-H2 | 15.91 | 12.55 | 48.76 | 72.19 | 56.88 | 57.77 | 71.33 | 39.07 | 61.55 |
| Phalanx+Attn (1:1) | 18.66 | 18.45 | 43.35 | 70.08 | 50.83 | 53.91 | 67.59 | 34.64 | 62.22 |
| Phalanx-multihybrid | 16.34 | 14.71 | 46.23 | 71.71 | 54.41 | 56.20 | 71.21 | 38.65 | 60.28 |

Table E.1: Zero-shot evaluation at 1.3B parameters / 100B tokens on FineWeb-Edu.

**Reading the table.** Phalanx+Attn (1:1) sits in the Transformer++ tier on every metric: within $\leq$ 1 pp on seven of the nine columns, with mild wins on HellaSwag, WinoGrande, LAMBADA accuracy, and BoolQ, and small losses on ARC-easy and ARC-challenge; both Wikitext and LAMBADA perplexities track Transformer++ within 0.13. Phalanx-multihybrid reaches the recurrent / hybrid baseline range: it ties with Mamba-2 on Wikitext perplexity, ARC-challenge, WinoGrande, and BoolQ; ties with Gated DeltaNet on ARC-easy, ARC-challenge, and BoolQ; and lands within 0.3–4 pp of GDN-H1 on the remaining metrics.

## F  Per-token-position perplexity at 1.3B

Complement to Section 6. We follow the per-position protocol of von Oswald et al. (2026): for each window length $L$ we stream random WikiText-103 chunks through Phalanx+Attn (1:1) and Phalanx-multihybrid, then compute mean cross-entropy at every position $p \in [0, L)$. We evaluate at 4K, 8K, and 16K context lengths.

Figure F.1 shows the per-position curves on linear and log-spaced position axes. We use $n = 82/41/20$ chunks at $L = 4K/8K/16K$.

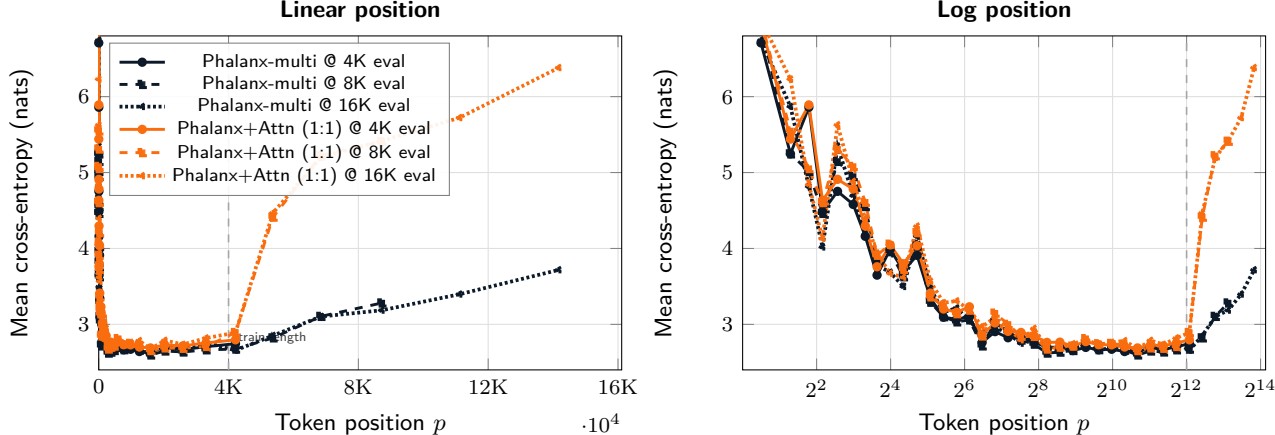

Figure F.1: Per-token-position cross-entropy at 1.3B / 100B FineWeb-Edu on WikiText-103 chunks.

