# OpenReview forum: "Sliding Window Recurrences for Sequence Models"
_TMLR — Accepted by TMLR_

### Review · Reviewer_fw2i · 2026-03-19

**Summary Of Contributions:**

This paper brings hardware-awareness to sub-quadratic alternatives for attention: previous works achieve the theoretical efficiency gains while maintaining competitive performance metrics when employed in hybrid architectures, but do not consider GPU implementation costs directly in their design. After extensive discussion of matrix theory of linear recurrences, the authors propose Sliding Window Recurrences, a family of sequence mixing primitives with a jagged window structure that emphasizes hardware-aligned data locality. These are instantiated via the proposed Phalanx layer implementation for empirical validation. Phalanx hybrid models match the standard transformer baseline and other hybrids in perplexity while training significantly faster.

**Additional Comments:**

I don't have much expertise in hardware and kernel implementation, so my understanding and analysis of those aspects of Sections 3-4 are rather surface-level, focusing more on consistency and clarity rather than deep correctness.

**Audience:**

Yes

**Audience Explanation:**

The intersection of efficient algorithms and hardware-aware deep learning is of high and broad interest to TMLR's audience. The matrix-theoretic perspective on parallel scans is a genuine and under-explored contribution that connects classical high-performance computing results to modern ML systems. Practitioners building hybrid architectures will find the Phalanx layer and B2P algorithm directly useful. Hybrid models are becoming a dominant paradigm in language modeling, and principled design of local operators within these hybrids is an open problem.

**Broader Impact Concerns:**

No broader impact statement was included.

**Claims And Evidence:**

Yes

**Claims Explanation:**

The throughput and perplexity results for Phalanx hybrid models are consistent and well-ablated. The mathematical derivations in Sections 3 and 4 seem rigorous, and the algorithm derivations map cleanly to their GPU implementations. However, some aspects of the experimental validation are limited:

* Evaluations are restricted to 1.3B parameter models. The claim that Phalanx-hybrid is a drop-in replacement for SWA is compelling at this scale, but it is unclear whether the quality and throughput trade-offs hold at other model sizes, larger or smaller. Scaling behavior of the jagged window is not studied.
* Perplexity (across the entire sequence) alone is an incomplete proxy for downstream quality. The paper does not include evaluations on established downstream benchmarks (e.g., LAMBADA, HellaSwag). Since Phalanx layers change the effective context length behavior, evaluations on tasks requiring multi-hop or long-range recall are crucial. Even analyzing the perplexity across the sequence length is a useful analysis.
* Some hyperparameters are explored experimentally, but the learning rate seems to be randomly selected (or taken from previous works) and constant across architectures, which could confound your claimed performance results if it happens to be a more optimal value for some architectures and less so for others.
* The ablation studies (Subsection "Comparing Phalanx variants") is a good start but could be further improved for consistency. Some are run for 40B tokens and others for 10B tokens. All seem to only be run on a single random seed and single learning. The results reporting is also inconsistent: one comparison is referred to an Appendix, another is included inline, and the last has no quantitative results reported. How do you know the gap between Phalanx and Phalanx-fixed=
* Figure 6.2: particularly for this bar chart that is serving to compare relative reductions in wall-clock time, not using a horizontal axis starting at 0 is misleading: upon first glance, the reader sees much larger relative reductions in wall-clock time than are actually indicated by the values shown. Either indicate the non-zero start using an axis break symbol or reset the axis to begin at 0.

**Requested Changes:**

At minimum, please include a table of downstream evaluation results and/or perplexity per token position (see Figure 3 of von Oswald et al., 2026). Also, fix Figure 6.2 as mentioned and include all ablation results, preferably with a consistent number of tokens. If compute allows, please include one other architecture size (8B, 3B, or even 400M) and run all architectures over a sweep of different learning rates, selecting the run with the best evaluation perplexity per architecture. Also, rerun the ablation studies with across different initialization seeds, keeping the data order the same.

Small typos:
* Section 5.2: "...have been explored for recurrence layers Dao (2024)" should be `\citep`.
* Section 6: "...while Phalanx layers have a head size of 16 and heads for k and v" seems to be missing the number of heads?
* Section 6: "...use the Flash Attention Dao (2023)" should be `\citep`.
* Section 6: Sequence length 32768 is mentioned in the text but not included in any results.
* Section 6: "...over these alternatives C, ..." -> "...over these alternatives (Appendix C), ..."


Johannes von Oswald, Nino Scherrer, et al. "MesaNet: Sequence modeling by locally optimal test-time training." ICLR 2026.

---

> ### Author Response · Authors · 2026-04-27
> **Response to Reviewer fw2i**
>
> We thank the reviewer for the careful and constructive review. The revision
> addresses the requested empirical checks, fixes the presentation issue in the
> time-to-target figure, and incorporates the listed corrections. We have also narrowed the
> paper's claims so that Phalanx is presented as a hardware-aware local mixer for
> hybrid language models, validated in the model scales and training settings we
> study, rather than as a general replacement across all sequence-modeling
> regimes.
>
> ---
>
> ## Downstream evaluation
>
> > *"At minimum, please include a table of downstream evaluation results and/or
> > perplexity per token position (see Figure 3 of von Oswald et al., 2026)."*
>
> We added two validation checks beyond training perplexity. First, a new 1.3B /
> 100B-token zero-shot table follows the Yang et al. (2024) suite: Wikitext,
> LAMBADA, PIQA, HellaSwag, WinoGrande, ARC-easy, ARC-challenge, and BoolQ. Phalanx+Attn (1:1) remains in the Transformer++ range, while Phalanx-multihybrid reaches the Mamba-2 / Gated DeltaNet hybrid range on
> several metrics. Second, a per-token-position perplexity appendix evaluates behaviour up to
> 4x the training context.
>
> ## Time-to-target figure
>
> > *"Either indicate the non-zero start using an axis break symbol or reset the
> > axis to begin at 0."*
>
> We fixed the figure by explicitly marking the non-zero horizontal-axis start
> with an axis break. The revised plot no longer relies on an unmarked truncated
> axis, so the wall-clock reductions are not visually overstated.
>
> ## Additional scale and learning-rate sweep
>
> > *"If compute allows, please include one other architecture size (8B, 3B, or
> > even 400M) and run all architectures over a sweep of different learning
> > rates, selecting the run with the best evaluation perplexity per
> > architecture."*
>
> The revision adds two checks in a new Section 6.1. First, we include a
> 150M / 5B-token architecture sweep on Dolma-3, comparing Transformer++,
> Mamba-2, Mamba-2+Attn, Gated DeltaNet+Attn, Phalanx+Attn variants, and
> Phalanx multihybrid. This adds both a smaller model scale and a different
> training dataset. The qualitative result is consistent with the 1.3B setting:
> Phalanx hybrids remain competitive in quality while improving training
> throughput.
>
> Second, we add a 340M / 15B-token learning-rate sweep for Phalanx+Attn (1:1)
> and Transformer++. We sweep peak learning rates in
> $\{1,2,4,8\}\times 10^{-4}$ and $1.6\times 10^{-3}$. Across the sweep,
> Phalanx+Attn reaches lower train and validation loss than Transformer++ at
> matched learning rates. This directly addresses the possibility that the
> reported gap was caused by a learning-rate choice favorable to one
> architecture.
>
> ## Ablation consistency and seeds
>
> > *"Also, rerun the ablation studies with across different initialization
> > seeds, keeping the data order the same."*
>
> > *"The ablation studies (Subsection 'Comparing Phalanx variants') is a good
> > start but could be further improved for consistency. Some are run for 40B
> > tokens and others for 10B tokens. [...]"*
>
> We agree that the original ablation presentation mixed exploratory and
> load-bearing comparisons. The revision separates these roles. The early
> 10B/40B runs are now framed as design exploration, while the main
> architectural comparisons are reported under matched recipes at the scales
> introduced above, including the 1.3B / 100B-token Phalanx+Attn and Phalanx
> multihybrid runs.
>
> We did not run a full multi-seed replication at 1.3B / 100B tokens. We instead
> scope the claims to the studied settings and add the learning-rate sweep above
> to reduce dependence on a single hyperparameter choice.
>
> ## Corrections and citation
>
> All five typos / inaccuracies flagged in the review are fixed:
>
> - the citation commands in Sections 5.2 and 6;
> - the missing K/V head count in Section 6;
> - the unsupported 32768-token statement;
> - the appendix reference typo;
> - the MesaNet citation in the related-work discussion.
>
> ## Broader impact
>
> > *"No broader impact statement was included."*
>
> We added a brief broader-impact statement. The revision states that Phalanx
> reduces the wall-clock cost of pretraining hybrid language models at a given
> quality target, reducing energy use and hardware time for a fixed experimental
> objective. We also note that training-efficiency improvements can shift the cost
> frontier for language-model development, so their impact depends on the
> downstream systems in which they are incorporated.

---

> > ### Comment · Reviewer_fw2i · 2026-05-06
> > **Response to paper updates**
> >
> > Thank you for your thorough responses and updates to the paper: all of my concerns were well-addressed.
> >
> > The tables of downstream evaluations (Tables D.1 and E.1) are useful for comparing the performance of the Phalanx layer to other model types. To make the tables even clearer to parse, you can add colors that represent the relative value per column, like the tables shown in Allen-Zhu (2025). You can also add some textual description of notable evals, such as which ones are factual recall, reasoning, etc. to help the readers understand the implications of the performance differences between model types.
> >
> > A small note, the learning rate sweep (Figure 6.5) suggests trying further higher learning rates until both models are clearly not improving. Also, we care less about "matched" learning rate performance and rather the best performing model of each type across the whole sweep.
> >
> > Zeyuan Allen-Zhu. "Physics of Language Models: Part 4.1, Architecture design and the magic of Canon layers." NeurIPS 2025.

---

### Review · Reviewer_7iRr · 2026-04-06

**Summary Of Contributions:**

This paper proposes Sliding Window Recurrences (SWR), a hardware-oriented truncation of linear recurrences for sequence modeling. The main idea is to rewrite recurrence computation in a hierarchical form, then aggressively truncate inter-block communication so that the resulting operator has a jagged local structure that maps well to GPU execution. The method is instantiated as a Phalanx layer used inside hybrid language models.

The paper is technically competent and generally clearly written. However, I find the contribution much stronger as a hardware-aware local-mixer design than as an algorithmic advance in sequence modeling. The central trade-off is driven mainly by communication locality and GPU alignment, rather than by a convincing argument that this is the right modeling bias. As a result, the practical value of the proposed trade-off remains unclear to me, especially beyond the specific hybrid training setting studied here. The paper explicitly frames SWR around “hardware-aligned data locality,” uses an aggressive block-size-16 truncation, and positions Phalanx as a local specialist while attention handles global context.

**Audience:**

Yes

**Audience Explanation:**

Readers working on frontiers of efficient sequence modeling methods may find the operator view and the implementation perspective interesting. However, in its current form, I do not think the paper provides sufficiently strong or general takeaways for a broader audience. The main message is tied to a fairly specific engineering trade-off, and the current evidence is too narrow to make that trade-off compelling beyond the setting studied here. The paper’s own positioning is largely as a local mixer for hybrid architectures, rather than as a more generally useful sequence-modeling mechanism.

**Claims And Evidence:**

No

**Claims Explanation:**

The paper shows that a hardware-oriented local recurrence can be useful as a component in hybrid LMs, but it does not convincingly establish that this is a broadly compelling algorithmic direction.

My main concern is that the key approximation is motivated primarily by hardware locality. The paper explicitly adopts an aggressive truncation with block size 16 to match GPU warp execution, and Phalanx is positioned as a local specialist while attention handles long-range dependencies. This makes the work more of a systems/operator co-design result than a novel contribution on sequence modeling. The empirical evidence is also limited for the scope of the claims. The main evaluation is perplexity plus training throughput on FineWeb-Edu. This is not enough to show robustness on tasks where long-range retrieval, reasoning, or downstream transfer matter more directly.

Finally, the comparisons and reporting are not fully convincing. The paper includes Transformer/SWA hybrids and a Mamba-2 throughput comparison, but not a sufficiently complete quality-throughput comparison against stronger recent efficient sequence models. Important details are also missing or too light: optimizer specifics, profiling details, memory footprint, and more systematic ablations on block size, truncation extent, and precision.

**Requested Changes:**

My concerns are mainly about the scope and motivation of the work. The central trade-off is hardware-motivated, but its algorithmic value remains unclear. The empirical case is narrow, focusing mostly on pretrain loss and training throughput on certain accelerators, and does not sufficiently address scalability and generalization abilities, downstream-task utility, or inference-time practicality. Comparisons to frontier efficient sequence models are also incomplete, and several important experimental details and ablations are still missing.

---

> ### Author Response · Authors · 2026-04-27
> **(1/2) Response to Reviewer 7iRr**
>
> We thank the reviewer for the detailed critique. The revision addresses the
> central concern by sharpening the paper's scope and strengthening the empirical
> case within that scope. We now present Phalanx as a windowed recurrence training
> mode for hybrid language models, not as a general sequence modeling replacement.
> Within that claim, we add broader comparisons, downstream evaluations,
> learning rate controls, and more complete implementation details.
>
> ---
>
> ## Scope and contribution
>
> > *"I find the contribution much stronger as a hardware-aware local-mixer design
> > than as an algorithmic advance in sequence modeling."*
>
> We agree that the right scope is not a claim that Phalanx replaces general
> sequence modeling mechanisms. The more precise claim is about the recurrence
> side of hybrid language models: attention already has a well established
> local/global split, with sliding window attention serving as the local variant
> and full attention serving as the global routing mechanism. Comparable windowed
> training modes for linear recurrences and SSM-style layers are much less
> developed, even though their transfer operators have structure that makes local
> truncation natural to study.
>
> The contribution is therefore a windowed recurrence formulation for this slot,
> not merely another local operator. The algorithmic content is the hierarchical
> transfer operator decomposition, the jagged truncation induced by that
> decomposition, and the Block Two-Pass algorithm that converts the resulting
> block bidiagonal recurrence operator into an efficient GPU training primitive.
> This positions the work at the algorithmic, systems, and numerical level: a
> hardware aware training mode for recurrence/SSM layers inside hybrid LMs.
>
> ## Modeling motivation
>
> > *"The central trade-off is driven mainly by communication locality and GPU
> > alignment, rather than by a convincing argument that this is the right
> > modeling bias."*
>
> We do not claim that windowed recurrence is the universally correct modeling
> bias for all SSMs. Linear recurrences span multiple regimes. When the
> coefficients satisfy $a_i \approx 1$ over long ranges, the layer approaches a
> linear attention style global accumulation regime. When some coefficients satisfy
> $|a_i| < 1$, the transfer coefficient from $u_j$ to $x_i$ contains the
> product $a_i a_{i-1}\cdots a_{j+1}$, so influence decays with distance. This
> decaying regime is especially relevant for low precision training and hardware
> execution, where sufficiently small transfer products eventually fall below the
> numerically meaningful scale of the computation.
>
> Our motivation is therefore not that all recurrent layers should be local.
> Rather, Phalanx studies a specific efficiency regime for recurrence/SSM layers
> in hybrids. Since global attention layers remain present to model long range
> interactions, it is natural to ask for the recurrence analogue of
> sliding window attention: a local training mode that preserves the useful
> near field recurrence dynamics while avoiding global carrier synchronization.
> The jagged approximation is the hardware aware form of this idea. It keeps the
> exact within block recurrence dynamics and nearest neighbor block coupling, but
> removes the global carrier solve induced by the hierarchical decomposition.
>
> The 16 token block is the smallest efficient tile used by our implementation,
> so it is also the most aggressive practical truncation we evaluate. This choice
> exposes the speed and quality tradeoff directly. We expanded the discussion of this
> design choice, along with uniform versus jagged truncation and head wise
> parameter sharing.
>
> ## Empirical breadth
>
> > *"The empirical evidence is also limited for the scope of the claims. The main
> > evaluation is perplexity plus training throughput on FineWeb-Edu."*
>
> The revision adds three empirical checks.
>
> First, we added a 150M / 5B token architecture sweep on Dolma-3, comparing
> Transformer++, Mamba-2, Mamba-2+Attn, Gated DeltaNet+Attn, Phalanx+Attn
> variants, and Phalanx multihybrid. This adds both a second dataset and a
> smaller model scale. Phalanx hybrids remain close to the Mamba-2 / Gated
> DeltaNet cluster in quality while training faster on H100.
>
> Second, we added a 340M / 15B token learning rate sweep for Phalanx+Attn (1:1)
> and Transformer++. Across the tested peak learning rates, Phalanx+Attn reaches
> lower train and validation loss at matched learning rates. This checks that the
> comparison is not explained by a learning rate choice favorable to one model.
>
> Third, we added a 1.3B zero shot downstream evaluation table following the
> benchmark suite used by Yang et al. (2024): Wikitext, LAMBADA, PIQA, HellaSwag,
> WinoGrande, ARC-easy, ARC-challenge, and BoolQ. Phalanx+Attn (1:1) lands in the
> Transformer++ range on this suite, while Phalanx multihybrid reaches the
> Mamba-2 / Gated DeltaNet hybrid range on several metrics.

---

> > ### Author Response · Authors · 2026-04-27
> > **(2/2) Response to Reviewer 7iRr**
> >
> > ## Comparisons to efficient sequence models
> >
> > > *"Comparisons to frontier efficient sequence models are also incomplete."*
> >
> > We expanded the comparison set in the added ablations. The 150M architecture
> > sweep includes Mamba-2 and Gated DeltaNet hybrid baselines under a matched
> > training setup. The 1.3B downstream table reports Phalanx rows against the
> > published Gated DeltaNet evaluation table, including RetNet, HGRN2, Mamba,
> > Mamba-2, DeltaNet, Gated DeltaNet, Transformer++, Samba, and GDN hybrids. This
> > places Phalanx against the relevant published efficient sequence baselines
> > without changing the paper's scoped claim.
> >
> > ## Practicality and reporting details
> >
> > > *"Important details are also missing or too light: optimizer specifics,
> > > profiling details, memory footprint, and more systematic ablations on block
> > > size, truncation extent, and precision."*
> >
> > We expanded the experimental and implementation reporting. The revised
> > experiments specify the tokenizer, model shape, optimizer, learning rate
> > schedule, batch size, sequence length, attention backends, and training setup.
> > The H100 throughput figure now reports training throughput, speedup, and peak
> > memory across sequence lengths.
> >
> > We also added an inference time practicality discussion. Phalanx recurrence
> > slots decode from compact recurrent state rather than a context growing
> > attention KV cache, while global attention layers in the hybrid stack retain
> > their usual cache and retrieval role. During prefill, B2P avoids the global
> > carrier solve by evaluating dense local tiles plus the nearest neighbor
> > correction.
> >
> > For the design ablations, the revision distinguishes between algorithmic design
> > choices and kernel dependent choices. We report the tested recurrence/gating
> > ablations and head sharing choices, and we explain why a block size sweep is
> > not a configuration only comparison: block size is tied to the Tensor Core tile
> > shape and would require matched kernels to be meaningful.

---

### Review · Reviewer_e7Xn · 2026-04-09

**Summary Of Contributions:**

This paper introduces Sliding Window Recurrences (SWR), a new class of sequence mixing primitives that explicitly align recurrence computations with GPU memory hierarchies. The key idea is to truncate linear recurrences into hardware-friendly “windows,” yielding a jagged, block-structured dependency pattern that reduces communication overhead. Building on a matrix-theoretic formulation of recurrences, the author(s) derived hierarchical algorithms and proposed the Block Two-Pass (B2P) algorithm for efficient implementation. They further introduced the Phalanx layer, a drop-in replacement for local attention or recurrence modules in hybrid architectures. Empirically, Phalanx achieves comparable perplexity to Transformer-based baselines while improving training throughput.

**Audience:**

Yes

**Audience Explanation:**

The paper targets a core intersection of interest in TMLR: sequence modeling, efficient architectures, and hardware-aware algorithm design.

**Claims And Evidence:**

Yes

**Claims Explanation:**

The paper provides a strong combination of theoretical development and empirical validation. The matrix formulation of linear recurrences and hierarchical decomposition is rigorous and clearly connected to algorithm design. The derivation of SWR and the B2P algorithm is well-motivated by both numerical properties (exponential decay) and hardware considerations.

On the empirical side, evaluating on large-scale language modeling is fairly convincing. I have a few comments; please see the requested changes section.

**Requested Changes:**

1. I find the paper exceptionally well-presented. Still, the paper would benefit from a bit more intuition in some of the denser technical sections, especially Section 3’s transfer-operator formulation, the flat vs hierarchical distinction, and Theorems 3.1-3.3 on the block/rank-one structure. In particular, it would help to more explicitly state the main conceptual takeaway that cross-block dependencies are compressed through a scalar carrier, which is what enables the efficient hierarchical algorithm. Similarly, in Section 4, the motivation for the jagged-window truncation could be explained more plainly as preserving exact local dynamics and nearest-neighbor coupling while removing global synchronization.

2. The paper currently presents its empirical results in a fairly specific setting: large-scale language modeling with a particular family of hybrid architectures, a fixed training recipe, and a limited set of baselines. Within that scope, the evidence is solid. However, several statements in the paper are phrased more broadly, as if the proposed approach were established as a generally effective design principle for sequence models or as a broadly applicable replacement across architectures and tasks.

3. The paper would benefit from a deeper analysis of key design choices such as block size, truncation strategy (uniform vs. jagged), and parameter sharing.

---

> ### Author Response · Authors · 2026-04-27
> **Response to Reviewer e7Xn**
>
> We thank the reviewer for the careful and encouraging review. The revision
> addresses the three main points raised in the requested changes: adding more
> intuition to the technical development, narrowing the empirical claims to match
> the evidence, and expanding the discussion of key design choices.
>
> ---
>
> ## Technical intuition and jagged-window motivation
>
> > *"The paper would benefit from a bit more intuition in some of the denser
> > technical sections, especially Section 3's transfer-operator formulation, the
> > flat vs hierarchical distinction, and Theorems 3.1-3.3 on the block/rank-one
> > structure."*
>
> We revised the technical exposition to make the carrier interpretation explicit:
> intra-block dynamics are solved exactly, while cross-block influence is mediated
> by a scalar carrier, yielding rank-one off-diagonal blocks. This also clarifies
> the flat-versus-hierarchical distinction. Flat scan factorizations reduce
> arithmetic depth but retain global communication patterns; the hierarchical view
> keeps dense local solves inside tiles and exchanges only the compact carrier
> across tiles.
>
> > *"Similarly, in Section 4, the motivation for the jagged-window truncation
> > could be explained more plainly as preserving exact local dynamics and
> > nearest-neighbor coupling while removing global synchronization."*
>
> The jagged approximation now follows directly from this carrier view: it keeps
> exact local recurrence dynamics and nearest-neighbor coupling, but removes the
> global carrier solve. This gives the Block Two-Pass algorithm, with one local
> solve per block followed by a nearest-neighbor reconstruction step.
>
> ## Scope of claims
>
> > *"Several statements in the paper are phrased more broadly, as if the proposed
> > approach were established as a generally effective design principle for
> > sequence models or as a broadly applicable replacement across architectures
> > and tasks."*
>
> We agree with the reviewer's reading and tightened the wording around the scope
> that was already central to the submission. The object of study is Sliding
> Window Recurrences: a windowed training mode for recurrence/SSM layers in hybrid
> language models. In this setting, attention remains the global routing component,
> while SWR focuses on the recurrent side, preserving exact within-block dynamics
> and nearest-neighbor coupling while avoiding the global carrier solve.
>
> To support this narrowed claim, the revision adds a 150M architecture sweep, a
> 340M learning-rate sweep, and a 1.3B zero-shot downstream evaluation table. The
> added experiments test the result across scale, dataset, and learning-rate
> choice while staying within the hybrid-LM setting that the method is designed
> for.
>
> ## Design choices
>
> > *"The paper would benefit from a deeper analysis of key design choices such as
> > block size, truncation strategy (uniform vs. jagged), and parameter sharing."*
>
> We expanded the design discussion around these choices.
>
> - **Block size.** The 16-token block is the smallest efficient tile used by our
>   implementation and therefore the most aggressive practical truncation we
>   evaluate. It exposes the speed-quality trade-off directly; changing it would
>   require a separately optimized kernel.
> - **Uniform vs. jagged truncation.** A uniform band at comparable radius leaves
>   many tile entries masked, underutilizing the GPU. The jagged window keeps
>   local tiles dense and adds only the rank-one nearest-neighbor tile.
> - **Parameter sharing.** The paper already explains this design choice in
>   Sections 3 and 4: sharing recurrence coefficients within a head is the
>   condition that turns the local recurrence from many independent matrix-vector
>   recurrences into one dense \(\ell \times d\) matrix-matrix tile solve. This is
>   what lets the B2P local solve target Tensor Cores rather than bandwidth-limited
>   per-channel scans, while still preserving independent channel states inside
>   each head. The same head-wise sharing principle appears in efficient
>   recurrence practice such as Mamba-2, and the Gated DeltaNet line is cited as
>   the surrounding design space for gated linear recurrence parametrizations.

---

### Author Response · Authors · 2026-04-23
**Request to hold final recommendations until revised PDF is posted**

Dear Action Editor and Reviewers,

Thank you for the thoughtful reviews. We are in the final stages of preparing our revision and per-reviewer responses to Reviewers e7Xn, 7iRr, and fw2i. The remaining work covers the items reviewers explicitly asked for: the 1.3B / 100B FineWeb-Edu headline run with the full downstream-evaluation table (Reviewers 7iRr, fw2i), the updated throughput / memory tables (Reviewer 7iRr), and additional revisions addressing the remaining reviewer comments.

We expect to upload the revised PDF and the author responses by **end of day 26 April 2026**. We would be grateful if the reviewers could hold off on submitting final recommendations until then, so that the response is in hand before any decision is finalized.

Thank you all for your patience.

Best,
The authors

---

> ### Author Response · Authors · 2026-04-27
>
> Dear Action Editor and Reviewers,
>
> Thank you again for your patience. We have now uploaded the revised PDF and posted the per-reviewer author responses addressing Reviewers e7Xn, 7iRr, and fw2i.
>
> Best,
> The authors

---

### Decision · Action_Editor_j7UT · 2026-06-08

**Recommendation:** Accept as is

**Audience:**

Yes

**Audience Explanation:**

Recently, hybrid language models have become increasingly popular among the TMLR audience. The paper focuses on the timely problem of improving the training efficiency of such models while taking into account the key characteristics of modern hardware.

**Claims And Evidence:**

Yes

**Claims Explanation:**

The paper aims to improve the training throughput and efficiency of hybrid language models that rely on linear recurrence. Based on a rigorous analysis, the paper proposes sliding window recurrence (SWR), which is designed to suit GPU memory hierarchies. The paper then leverages SWR to design the Phalanx layer to replace other local sequence mixing layers in hybrid sequence modeling architectures.

While the initial submission had a much broader scope which was inadequately supported, the authors appropriately narrowed down the messaging of the paper based on the reviewers’ comments. The revision also provided additional results, including hyperparameter sweeps, experiments with different model sizes, and more comprehensive evaluation criteria such as per-position perplexity and downstream performance. In its current form, the submission clearly supports the main claim: SWR-based Phalanx layers can improve the training speed and quality of hybrid sequence models.